# Harnessing cholesterol uptake of malaria parasites for therapeutic applications

Merryn Fraser [ID] [1,2,5,6], Blake Curtis [ID] [3,4,6], Patrick Phillips[1], Patrick A Yates[3], Kwong Sum Lam [ID] [1], Otto Netzel[2], Giel G van Dooren [ID] [1], Alyssa Ingmundson [ID] [2], Kai Matuschewski [ID] [2], Malcolm D McLeod [ID] [3,7 ✉] & Alexander G Maier [ID] [1,7 ✉]

## Abstract

Parasites, such as the malaria parasite *P. falciparum*, are critically dependent on host nutrients. Interference with nutrient uptake can lead to parasite death and, therefore, serve as a successful treatment strategy. *P. falciparum* parasites cannot synthesise cholesterol, and instead source this lipid from the host. Here, we tested whether cholesterol uptake pathways could be 'hijacked' for optimal drug delivery to the intracellular parasite. We found that fluorescent cholesterol analogues were delivered from the extracellular environment to the intracellular parasite. We investigated the uptake and inhibitory effects of conjugate compounds, where proven antimalarial drugs (primaquine and artesunate) were attached to steroids that mimic the structure of cholesterol. These conjugated antimalarial drugs improved the inhibitory effects against multiple parasite lifecycle stages, multiple parasite species, and drug-resistant parasites, whilst also lowering the toxicity to human host cells. Steroids with introduced peroxides also displayed antimalarial activity. These results provide a proof-of-concept that cholesterol mimics can be developed as a drug delivery system against apicomplexan parasites with the potential to improve drug efficacy, increase therapeutic index, and defeat drug resistance.

**Key words** *Plasmodium falciparum*; malaria; cholesterol; drug-resistance; drug-delivery
**Subject Categories** Metabolism; Microbiology, Virology & Host Pathogen Interaction

## Introduction

Malaria is a vector-borne infectious disease caused by *Plasmodium* parasites (Maier et al, 2019). *Plasmodium falciparum* is the deadliest species, and has developed resistance to all antimalarial drugs currently in use, underscoring the need for new treatments (Haldar et al, 2018; WHO, 2023). *P. falciparum* blood stages can be cultured in vitro, allowing thorough investigations and high throughput screening to identify novel inhibitors (Smilkstein et al, 2004; Gamo et al, 2010; Guiguemde et al, 2010; Rottmann et al, 2010; Spangenberg et al, 2013). Drug intervention has traditionally focused on the asexual blood stage, since it is the stage that causes malaria disease. However, attention has now also turned to other stages of the parasite lifecycle, such as gametocyte and liver stages. Successfully targeting these stages can prevent the disease and block transmission aiding in the eradication of malaria (Alonso et al, 2011; Derbyshire et al, 2011; Birkholtz et al, 2022).

Primaquine and its analogue tafenoquine are the only drugs currently approved to target the dormant liver-stage forms of *P. vivax* and *P. ovale*, which can cause autologous relapses (Lell et al, 2000; Baird and Hoffman, 2004; Vale et al, 2009; Ashley et al, 2014; Lacerda et al, 2019). Primaquine is a pro-drug which is metabolised in the human body into activated forms, either by direct oxidation in RBCs or by the cytochrome P450 system in the liver (Fasinu et al, 2019; Camarda et al, 2019). Its efficacy against asexual intraerythrocytic *P. falciparum* stages is moderate, but primaquine also has activity against sexual stages, termed gametocytes (Rieckmann et al, 1969; Graves et al, 2014; Ashley et al, 2014; Camarda et al, 2019). However, primaquine has risks associated with toxicity in high doses, and its use is contraindicated for patients with glucose-6-phosphate dehydrogenase deficiencies since it can cause fatal haemolysis in these individuals (Vale et al, 2009; Ashley et al, 2014). Therefore, using medicinal chemistry to improve the efficacy and/or decrease toxicity of primaquine would be advantageous.

Artemisinin and its derivatives such as artesunate are the current front-line treatments for malaria (WHO 2023). However, their efficacy and usefulness are threatened by the emergence of drug resistant parasites in the field—a fate they share with all other commercially available antimalarial drugs. Hence, strategies to overcome drug resistance would be highly valuable tools in the fight against this devastating parasitic disease.

Cholesterol is a steroidal nutrient which serves as a vital membrane component of animal cells and some protozoa. Cholesterol consists of a tetracyclic hydrocarbon ring system, a saturated hydrocarbon side chain from C-17, and a hydroxy group from C-3 (Fig. 1A). Neither parasite nor its host red blood cell (RBC) can synthesise cholesterol

[1]Research School of Biology, The Australian National University, Canberra 2601, Australia. [2]Department of Molecular Parasitology, Institute of Biology, Humboldt University, Berlin 10115, Germany. [3]Research School of Chemistry, The Australian National University, Canberra 2601, Australia. [4]Metabolism of Microbial Pathogens, Robert Koch Institute, Berlin 13353, Germany. [5]Present address: Harvard T. H. Chan School of Public Health, Harvard University, Boston, MA 02115, USA. [6]These authors contributed equally: Merryn Fraser, Blake Curtis. [7]These authors jointly supervised this work: Malcolm D McLeod, Alexander G Maier. ✉E-mail: Malcolm.McLeod@anu.edu.au; Alex.Maier@anu.edu.au

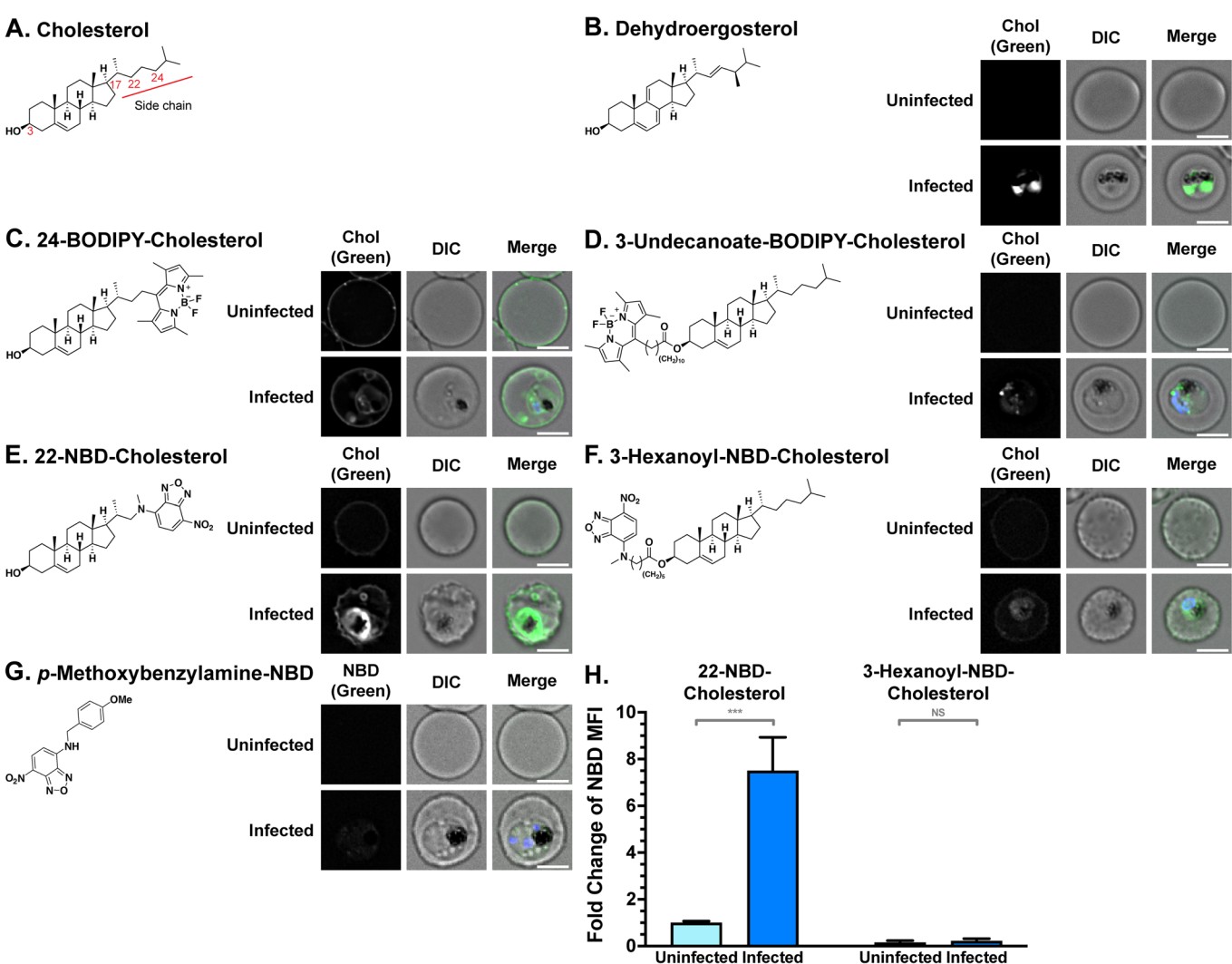

**Figure 1. Fluorescent cholesterol analogues are taken up by iRBCs, with preference for cholesterol side chain conjugates.**

(A) Structure of cholesterol, showing the numbering of key carbon atoms within the molecule. (B–F) Structure and representative subcellular fluorescence localisation of (B) dehydroergosterol, (C) 24-BODIPY-cholesterol, (D) 3-undecanoate-BODIPY-cholesterol, (E) 22-NBD-cholesterol (F) 3-hexanoyl-NBD-cholesterol and (G) p-methoxybenzylamie-NBD in uninfected red blood cells (uRBCs) and infected red blood cells (iRBCs,) visualised by deconvolution fluorescence microscopy after 24 h of incubation; BODIPY and NBD fluorescence (depicted in green) were detected at 475 nm (ex)/525 nm (em), and dehydroergosterol (depicted in green) and Hoechst 33342 fluorescence (parasite DNA; depicted in blue) were detected at 390 nm (ex)/435 nm (em). Scale bar = 4 μm. DIC: Differential Interference Contrast. Microscope settings were identical between uRBC/iRBC pairs for each fluorophore but changed between different fluorophores. Approximately 100 images were acquired for each condition. (H) Quantification of NBD-lipid fluorescence, measured by foldchange of NBD mean fluorescence intensity (MFI) of whole cells in flow cytometry. Note that greater uptake is observed when fluorophores are introduced on the cholesterol side chain rather than at the C-3 position. Shown are mean values (± S.D.) $n = 3$ independent experiments. NS, not significant; ***$p < 0.001$ (ANOVA). Source data are available online for this figure.

de novo, but this lipid is vital to parasite survival (Lauer et al, 2000; Labaied et al, 2011; Petersen et al, 2017; Hayakawa et al, 2020; Maier and van Ooij, 2022). Despite this lack of synthesis capacity, cholesterol accumulates in infected RBCs (iRBCs), particularly in late stage (trophozoite and schizont) iRBCs and gametocytes. Cholesterol storage in gametocytes might be an adaption important for transmission, where parasites encounter a low cholesterol environment in the mosquito (Tran et al, 2016; Ridgway et al, 2022). Cholesterol normally localises to the RBC membrane, but upon *Plasmodium* infection, cholesterol is depleted from the RBC membrane and accumulates in the intracellular parasite (Tran et al, 2016; Fraser et al, 2021), and interrupting this process leads to altered

cholesterol distribution and parasite death (Tokumasu et al, 2014; Istvan et al, 2019; Bhatnagar et al, 2019; Hayakawa et al, 2020). Furthermore, depletion of cholesterol from the RBC membrane using methyl-β-cyclodextrin can stop parasite development and prevent its invasion (Lauer et al, 2000; Haldar et al, 2002; Frankland et al, 2006; Ahiya et al, 2022). Clinical data indicate that lipoprotein-bound blood cholesterol is lower in malaria patients (Lambrecht et al, 1978; Chukwuocha and Eke, 2011; Orimadegun and Orimadegun, 2015; Visser et al, 2017; Rakovac Tisdall et al, 2018; Megabiaw et al, 2022), suggesting that *Plasmodium* parasites might act as a cholesterol sink. Together, these observations indicate that cholesterol moves from the RBC membrane and the extracellular environment into the parasite.

We hypothesised that conjugating cholesterol to antimalarial drugs could hijack existing cholesterol uptake pathways to maximize drug delivery to the parasite, potentially increasing drug efficacy.

# Results

## Parasitised red blood cells take up fluorescent cholesterol analogues

To gain insights into how cholesterol (Fig. 1A) gets into parasites, we tested several commercially available fluorescent cholesterol analogues for accumulation in intra-erythrocytic parasites. We started with the naturally occurring fungal sterol dehydroergosterol, which is structurally similar and mimics the properties of cholesterol, but exhibits intrinsic fluorescence due to the presence of additional double bonds (Fig. 1B) (McIntosh et al, 2008). We incubated RBC infected with ring stage parasites with dehydroergosterol for 24 h and imaged the resulting trophozoite infected red blood cells (iRBCs) using deconvolution fluorescence microscopy. Fluorescence was visible in iRBCs, with the fluorescence concentrated within the parasite, indicating uptake of the compound (Fig. 1B).

We next investigated the uptake of analogues where a fluorophore was conjugated from either the C-3, C-22, or C-24 position of cholesterol (Fig. 1C–F). As with dehydroergosterol, the ring-stage parasite culture was incubated with the cholesterol analogues for 24 h and imaged using deconvolution fluorescence microscopy. 24-BODIPY-cholesterol fluorescence (green) was visible in the RBC membrane of both uninfected RBCs (uRBCs) and iRBCs, and in the intracellular parasites inside erythrocytes (parasite DNA depicted in blue) (Fig. 1C). Some fluorescence also appeared to localise within the iRBC cytoplasm, possibly due to incorporation into parasite-induced membranous structures (27). Fluorescence of 3-undecanoate-BODIPY-cholesterol was faint and only visible in the parasite (Fig. 1D).

22-NBD-cholesterol fluorescence was bright around the area of the parasite in iRBCs and faintly visible in the RBC membrane of both uRBCs and iRBCs (Fig. 1E). 3-Hexanoyl-NBD-cholesterol fluorescence was faintly visible in the RBC membrane and in the parasite (Fig. 1F). An NBD molecule not conjugated to cholesterol also exhibited only faint fluorescence in uRBCs and iRBCs (Fig. 1G). The differences in fluorescence intensity between uRBCs and iRBCs were also quantified using flow cytometry. Uptake of 22-NBD-cholesterol was approximately 8-fold higher in iRBCs than uRBCs (Fig. 1E,H), while the amount of 3-hexanoyl-NBD-cholesterol taken up was not significantly different between iRBCs than uRBCs and was much lower than 22-NBD-cholesterol (Fig. 1F,H).

These results demonstrate that the addition of bulky substituents to the cholesterol backbone does not prevent the uptake into the parasite. This uptake into iRBCs was most notable, and greatly enhanced compared to uRBCs, when fluorescent groups were added to the cholesterol side chain, and the C-3 hydroxy group was unaltered (Fig. 1C–H).

## Coupling of the antimalarial compound primaquine to a steroid increases its efficacy and pharmacological profile against asexual *P. falciparum*

Conjugating compounds with antiplasmodial activity to cholesterol or cholesterol-like steroids may result in greater specificity by hijacking the cholesterol uptake pathways to deliver the drug to the parasite. We selected primaquine as a proof-of-principle compound since this compound could be improved to increase efficacy, therapeutic index, and reduce off-target effects. Primaquine is also an attractive candidate due to its activity against all lifecycle stages in the human host and in particular against transmissible gametocytes. It is unlikely to require cleavage from a carrier for it to be active and has a terminal amine, which facilitates the ease of conjugation. Dehydroepiandrosterone (DHEA) is a steroid which is structurally similar to cholesterol, with an identical hydrocarbon ring structure and hydroxy group at C-3. For ease of synthesis, we used DHEA as a chemical surrogate for cholesterol to create a steroid-primaquine conjugate where primaquine (Fig. 2A-i) is connected to the C-17 position via a succinate linker, naming the resulting compound C-17-PQ (Fig. 2A-ii). We attached primaquine via its terminal primary amine in the form of an amide to minimise any disturbance to function, since the active quinoline core is left unchanged. The steroid primaquine conjugate was prepared from DHEA in five steps with a 76% overall reaction yield. Synthetic schemes, experimental procedures, and characterisation for synthesised compounds can be found in the APPENDIX. We also conjugated primaquine to the cholesterol C-3 position using the same succinate linking strategy, forming the compound C-3-PQ (Fig. 2A-iii). Additionally, we synthesised several control compounds to mimic portions of the full conjugates: primaquine attached to a succinate linker (PQ-link) (Fig. 2A-iv), DHEA attached to a succinate linker from C-17 (C-17-link) (Fig. 2A-v), and cholesterol attached to a succinate linker on C-3 (C-3-link) (Fig. 2A-vi), with all succinate linkers containing a terminal isopropyl ester.

We tested these compounds for inhibitory effects on parasite growth over 24 h to mimic the conditions of the fluorescent cholesterol uptake assays, starting with ring-stage parasites. The steroid-coupled primaquine exhibited a ~ 4-fold lower 50% inhibitory concentration ($IC_{50}$) than primaquine alone (3.1 μM vs. 11.5 μM; $p = 0.006$) (Figs. 2B and EV1A). In addition, the dose–response curve is also steeper, indicative of different dynamics of the inhibitory action. Hence, the difference between the two compounds is more pronounced (~8-fold) when comparing $IC_{90}$ values (41 μM for primaquine vs. 5.0 μM for C-17-PQ; $p = 0.016$). In contrast to C-17-PQ, C-3-PQ showed no inhibition of parasite growth up to 100 μM. The control compound PQ-link showed a higher $IC_{50}$ than primaquine alone, supporting the notion that the addition of a steroid to primaquine, rather than the succinate linker alone, enhances its activity. C-17-link similarly does not show the same level of activity as C-17-PQ, indicating that both the drug and the steroid components are important for parasite inhibition. C-3-link showed no inhibition of parasite growth up to 200 μM.

In order to investigate the dynamics of the drug's effect, we also determined the $IC_{50}$ of the compounds with longer incubation times of 48 and 72 h. The $IC_{50}$ of C-17-PQ did not change between the different timepoints ($p > 0.05$ for all), indicating that the drug reached its maximum activity after 24 h. In contrast to C-17-PQ, the $IC_{50}$ of primaquine decreases with longer incubation periods (Fig. EV1A,B; $p = 0.026$ 24 h vs. 48 h; $p = 0.0024$ 24 h vs 72 h), providing further evidence that drug delivery is a limiting factor for primaquine, which can be overcome by coupling to a steroid. While the inhibitory curve for primaquine was steeper at 72 h compared

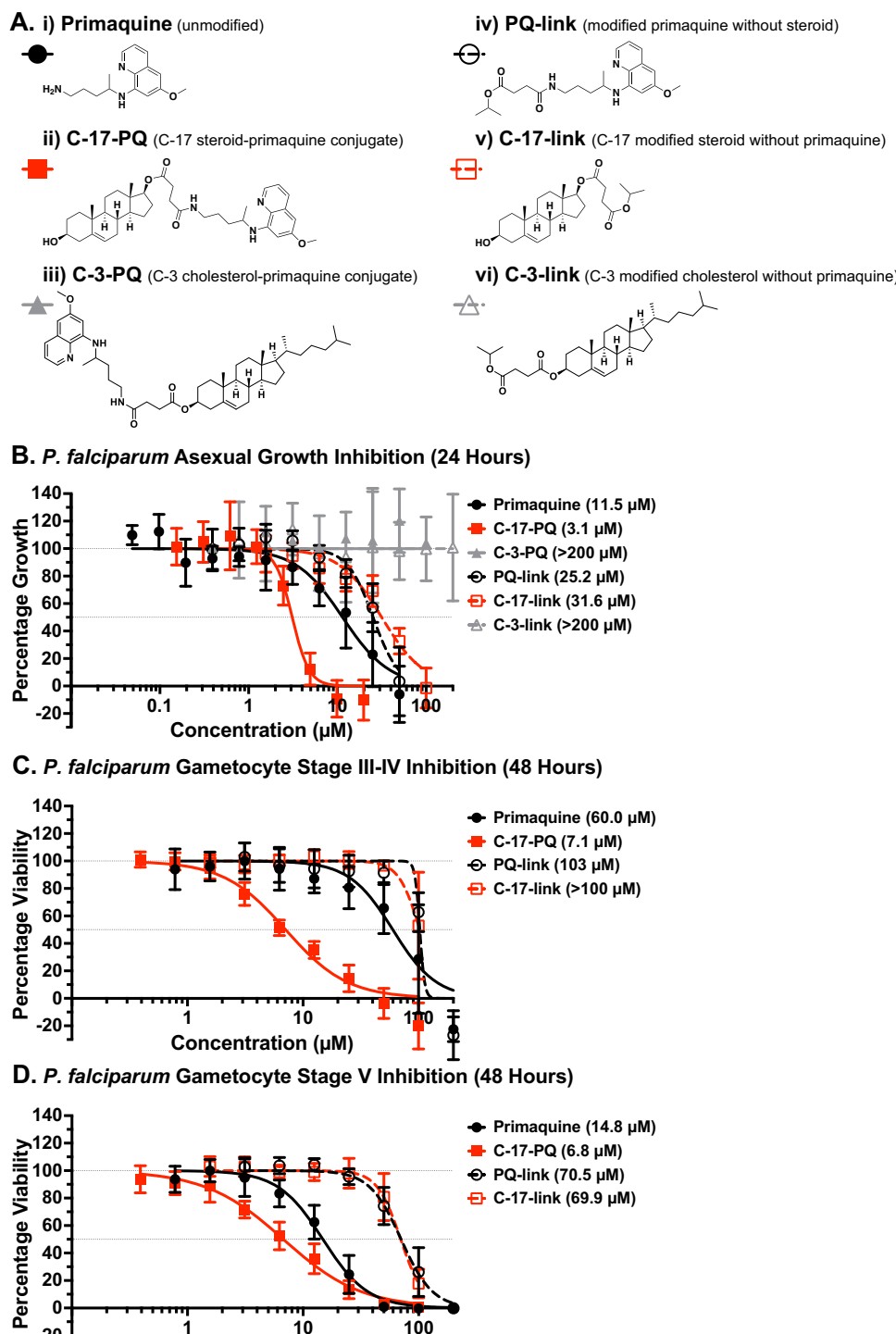

**Figure 2. Coupling of primaquine to a steroid increases its potency against *P. falciparum*.**

(A) Chemical structures of (i) primaquine; (ii) steroid with primaquine conjugated from C-17 (C-17-PQ), (iii) cholesterol with primaquine conjugated from C-3 (C-3-PQ) (iv) primaquine with a linker (PQ-link), (v) steroid molecule with a linker at C-17 (C-17-link), and (vi) cholesterol with a linker at C-3 (C-3-link). (B) Dose–response assay (24 h) of compounds against asexual *P. falciparum* growth (starting at ring stages). Shown are mean values (± S.D.). 50% inhibitory concentration ($IC_{50}$) is in brackets. $n = 3$ independent experiments. (C) Dose–response assay (48 h) against sexual (stage III–IV gametocytes) *P. falciparum* viability. $n = 3$ independent experiments. Shown are mean values (± S.D.). (D) Dose–response assay (48 h) against sexual (stage V gametocytes) *P. falciparum* viability. Shown are mean values (± S.D.). $IC_{50}$ is in brackets. $n = 3$–4 independent experiments. Source data are available online for this figure.

to the 24 h exposure time (Hill slope −1.48 and −1.65 for 72 and 24 h, respectively), it remained consistently flatter than the C-17-PQ curves at any timepoint (Hill slope −4.20 and −4.56 for 72 and 24 h, respectively). Accordingly, while the $IC_{50}$ values after 72 h exposure time for primaquine approaches the one for C-17-PQ, the $IC_{90}$ values still differ (primaquine, 11.4 μM; C-17-PQ, 4.0 μM). None of the control compounds showed significant differences in $IC_{50}$ between incubation times, nor did the C-3 coupled compounds (Fig. EV1C; $p > 0.05$).

The slower drug action of primaquine could be a notable shortcoming of this drug, whereas another antimalarial drug, chloroquine, reaches full activity after 24 h of exposure (Fig. EV1D). Hence, steroid conjugation may overcome the slow onset of primaquine activity. Together, these data suggest that the C-17 steroid-primaquine conjugate is a more potent and faster acting drug than primaquine alone against the growth of asexual *P. falciparum*, identifying modified C-17 steroids as promising candidates for efficient drug delivery into parasitised RBCs.

### Steroid-primaquine conjugation increased inhibition of *P. falciparum* gametocyte viability

Primaquine also exhibits gametocidal activity (Rieckmann et al, 1969; Graves et al, 2014; Ashley et al, 2014; Camarda et al, 2019). Notably, gametocytes accumulate large quantities of cholesterol as they develop through stages I–V (Tran et al, 2016; Ridgway et al, 2022). We therefore hypothesised that our steroid-primaquine conjugate C-17-PQ would show improved activity against the sexual blood stages of *P. falciparum*, and measured gametocyte viability (stage III–IV) with a live cell mitochondrial dye. C-17-PQ exhibited a 9-fold reduction in $IC_{50}$ compared to primaquine alone after 48 h of incubation, with an $IC_{50}$ of 6.9 μM (primaquine 63.5 μM; $p = 0.025$), indicating that the conjugate kills gametocytes more efficiently (Figs. 2C and EV1E). Both control compounds, PQ-link and C-17-link, showed a significantly higher $IC_{50}$ (>100 μM) than C-17-PQ ($p = 0.0017$ and $p = 0.023$ respectively), indicating that both drug and steroid components are required for enhanced activity.

Once gametocytes have reached stage V, they are transmissible and can persist in the blood stream for several weeks. Therefore, we also tested the activity of our conjugate against stage V gametocytes (Fig. 2D). C-17-PQ showed similar activity against this stage compared to the earlier stages, with an $IC_{50}$ of 6.8 μM. This $IC_{50}$ represents a >2-fold reduction compared to primaquine alone after 48 h of incubation. These data indicate that gametocyte inhibition was improved with the steroid conjugation strategy, with a more pronounced difference in earlier stages.

### The steroid-primaquine conjugate inhibits the growth of liver stage *P. berghei* more effectively than primaquine alone

Since primaquine is also used to target liver stage parasites, we wanted to investigate if the steroid-primaquine conjugate had any difference in efficacy against this stage. These experiments utilised luciferase-expressing sporozoites of the rodent parasite species *Plasmodium berghei* to infect human hepatoma cells (Huh7) (Figs. 3A and EV1F). The steroid-coupled primaquine had an $IC_{50}$ of 0.2 μM, exhibiting a ~15-fold reduction in $IC_{50}$ compared to primaquine alone (2.9 μM; $p = 0.038$) after 48 h of treatment

commencing from sporozoite inoculation. Both of the control compounds, PQ-link and C-17-link, showed a significantly higher $IC_{50}$ than C-17-PQ ($p = 0.016$ and $p = 0.004$, respectively).

To better understand the inhibitory activity of the steroid-primaquine conjugate C-17-PQ, we conducted fluorescence microscopy experiments to visualise parasite development inside the infected cells after inoculation with a constant number of sporozoites. We detected a clear effect on liver stage parasite development upon treatment with either primaquine or C-17-PQ (Fig. 3B). Quantification of parasite area (Fig. 3C) revealed a smaller parasite size when infected cells were treated with 0.3 μM C-17-PQ (48% reduction compared to untreated parasites; $p < 0.001$) or 3 μM C-17-PQ (90% reduction; $p < 0.001$). In contrast, primaquine alone resulted in smaller parasites only at 3 μM (61% reduction; $p < 0.001$), and not at 0.3 μM ($p = 0.78$). These differences between C-17-PQ and primaquine at the matching concentrations were statistically significant ($p = 0.0013$ for 3 μM; $p < 0.001$ for 0.3 μM). Similarly, the parasite number (per coverslip) was significantly lower when treated with C-17-PQ compared to the same concentration of primaquine ($p < 0.001$ for all), again indicating that the conjugate performed better than primaquine alone (Fig. 3D).

Together, these data indicate that liver stage inhibition was improved with the conjugation strategy, decreasing the parasite burden.

### The steroid-primaquine conjugate inhibits the growth of another apicomplexan parasite, *Toxoplasma gondii*

*Toxoplasma gondii* is an apicomplexan parasite that is related to *P. falciparum* and causes toxoplasmosis in humans and animals. *T. gondii* is less susceptible to primaquine than *Plasmodium* (Holfels et al, 1994; Radke et al, 2018), and this drug is not used for the treatment of toxoplasmosis. Like *Plasmodium*, *T. gondii* relies on the uptake of cholesterol to survive (Coppens et al, 2000, 2006; Bottova et al, 2009). As a proof-of-concept for enhanced uptake by conjugating a steroid to primaquine, we investigated whether the steroid-primaquine conjugate could inhibit the growth of *T. gondii* tachyzoites, the parasite stage which causes the disease (Figs. 3E and EV1G).

We found that the steroid-primaquine conjugate exhibited a significantly lower $IC_{50}$ than unconjugated primaquine against *T. gondii* (2.8 μM vs. 9.7 μM; ~3.5-fold difference; $p = 0.015$). Both primaquine and the steroid-primaquine conjugate were more potent than either of the control compounds, demonstrating that both the steroid and drug components are required to effectively inhibit growth.

### Steroid-primaquine conjugate shows lower cytotoxicity against human cells

When assessing a potential antiparasitic drug, it is essential to consider not only the effect on the parasite, but also the host cells, to determine the compound selectivity toward the parasite and to gauge a safe therapeutic window. Primaquine has a relative small therapeutic window, resulting in an extended treatment duration (Ashley et al, 2014). We therefore investigated whether the compounds were toxic to three selected cell types: human hepatoma cells (Fig. 3F), human fibroblasts (Fig. 3G), and human embryonic kidney cells (Fig. EV1H).

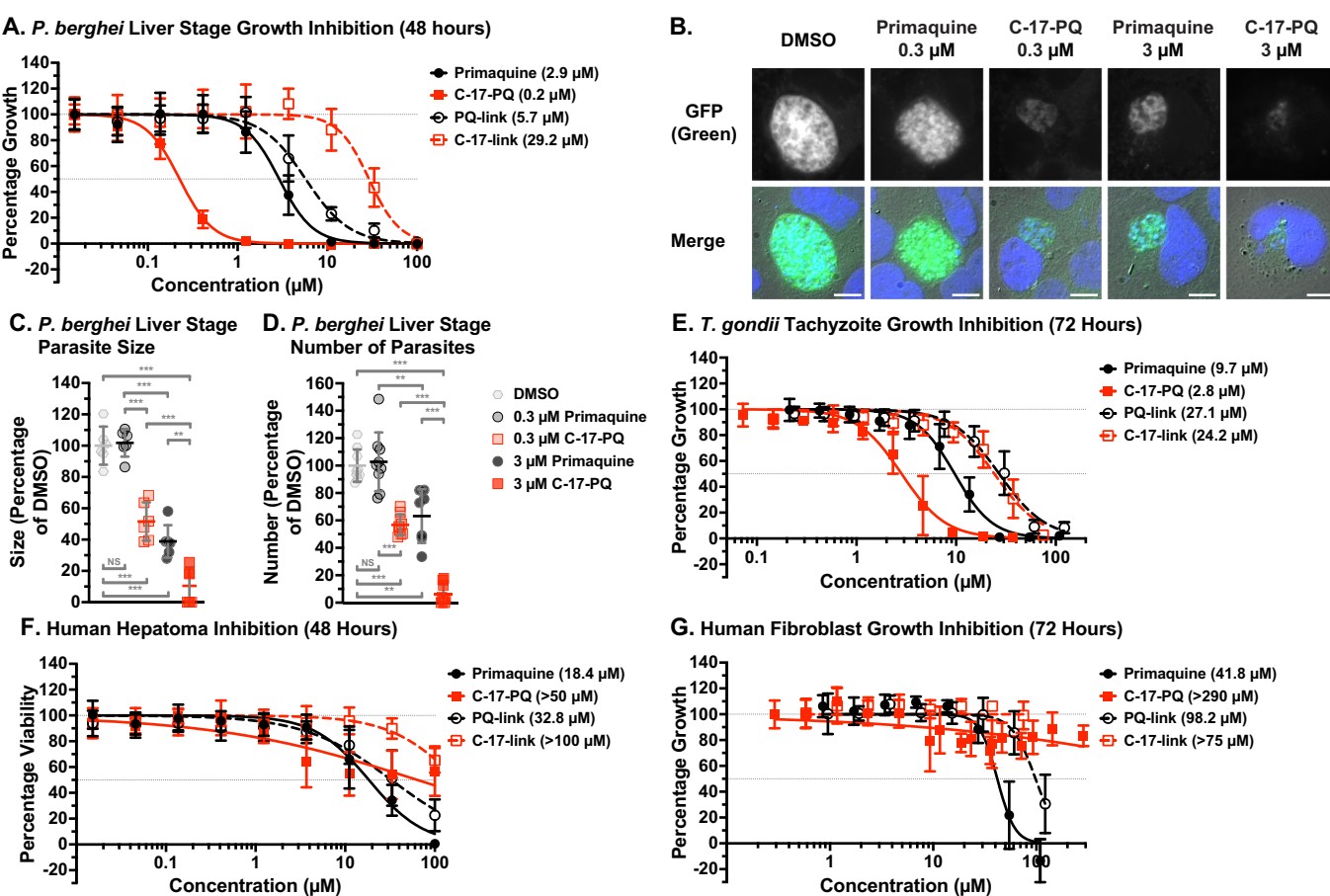

**Figure 3. Steroid conjugation enhances the inhibitory effect of primaquine against *P. berghei* liver stages and *Toxoplasma gondii* tachyzoites while having a lower inhibitory effect than primaquine on the viability of human hepatoma cells and fibroblasts.**

(A) Dose–response growth assay (48 h) against *P. berghei* incubated with primaquine, C-17-PQ, PQ-link, or C-17-link. Shown are mean values (± S.D.) IC50 is in brackets. $n = 3$ independent experiments. (B) *P. berghei* liver stage schizonts expressing GFP-Luc in human hepatoma cells (Huh7) (48 h post-infection) treated with 0.3 or 3 μM primaquine or C-17-PQ, or solvent control (DMSO), visualised by fluorescence microscopy. GFP fluorescence (depicted in green) was detected at 470 nm (ex)/525 nm (em) and DAPI fluorescence (DNA; depicted in blue) was detected at 359 nm (ex)/445 nm (em). Scale bar = 10 μm. Fluorescence intensity is not comparable between images. (C) Normalised mean size quantification ($n = 2$ independent experiments; mean of all parasites per coverslip) and (D) number of parasites per coverslip ($n = 3$ independent experiments) of liver stage parasites after inoculation with an equal number of sporozoites and treatment with 0.3 or 3 μM primaquine or C-17-PQ, or solvent control (DMSO). Shown are individual data points for each technical replicate, with centre bar representing mean values (± S.D.). At least 100 images were analysed for each condition. NS, not significant; **$p < 0.01$; ***$p < 0.001$ (ANOVA). (E) Dose–response growth assay (72 h) against *Toxoplasma gondii* tachyzoites incubated with primaquine, C-17-PQ, PQ-link, or C-17-link. Shown are mean values (± S.D.). IC50 is in brackets. $n = 4$ independent experiments. (F) Dose–response viability assay (48 h) against human hepatoma cells (Huh7) incubated with primaquine, C-17-PQ, PQ-link, or C-17-link. Shown are mean values (± S.D.). Calculated IC50 is in brackets. $n = 3$ independent experiments. (G) Dose–response growth assay (72 h) against human foreskin fibroblasts (HFF) incubated with primaquine, C-17-PQ, PQ-link, or C-17-link. Shown are mean values (± S.D.). IC50 is in brackets. $n = 3$ independent experiments. Source data are available online for this figure.

In human hepatoma cells (used for *P. berghei* liver stage infection), primaquine and PQ-link inhibited viability with an IC50 of 18.4 and 32.8 μM, respectively (Fig. 3F). In contrast, when treated with C-17-PQ, the viability of these cells did not drop below 50% over the concentration range tested (up to 100 μM). However, a gradual decline in viability was detected at higher concentrations. The control steroid-linker compound showed only modest cytotoxicity at high concentrations (>30 μM).

In human foreskin fibroblasts (HFF cells, used for *T. gondii* infection), primaquine and PQ-link inhibited fibroblast growth with an IC50 of 41.8 μM and 98.2 μM, respectively, while no notable inhibition was observed with C-17-link in the range of concentrations tested. Similar to the hepatoma cells, C-17-PQ

showed a slow, consistent decline in viability, but was not able to inhibit more than 30% of the fibroblast growth at the range of concentrations tested (Fig. 3G). Cytotoxicity against human embryonic kidney cells (HEK293) showed similar results, with C-17-PQ again showing a slow, gradual inhibition with an estimated IC50 well above that of primaquine or the other controls (Fig. EV1H). Primaquine and PQ-link showed IC50 values of 10.6 and 13.3 μM, respectively. C-17-link inhibited HEK293 cells with an IC50 of 43.6 μM.

Together, these data show that steroid-coupling can also enhance the effectiveness of primaquine in another apicomplexan parasite, *T. gondii*, whilst still showing low cytotoxicity in the human host cells.

## Steroid-artesunate conjugation overcomes drug resistance in *Plasmodium* ring stages

One of the largest challenges in malaria control today is the emergence and growing prevalence of parasite resistance to frontline treatments. Artemisinin combination therapies (ACTs) are the main antimalarial therapy in many parts of the world and often the drug of last resort, but emergence and spread of mutations in the parasite protein Kelch-13 results in delayed clearance and clinical treatment failures due to these mutations have been reported (Noedl et al, 2008; Dondorp et al, 2009; Ariey et al, 2013). The management of ACT-resistant parasites requires longer treatment times, which impacts on cost and compliance with the treatment regime, and increases the chance of patient morbidity and mortality before the infection can be cured (Dondorp et al, 2009; Noedl et al, 2008). We therefore investigated whether our conjugation strategy could increase the effectiveness of an artemisinin derivative, artesunate, against parasites resistant to this drug.

We synthesised a steroid-artesunate conjugate, compound C-17-ART, in two steps with a 31% overall yield from artesunate (see APPENDIX) by attaching the succinate group of artesunate to the C-17 position of DHEA (Fig. 4A). Similar to PQ-link, we also synthesised a control compound, ART-link, where a terminal isopropyl ester group was added to the succinate group of artesunate as a chemical linker. The steroid-linker conjugate, C-17-link, served as the other control.

We tested these compounds in a ring stage survival assay, which allows the detection of resistant parasites and is the standard method for investigating resistance to artemisinin and its derivatives (Witkowski et al, 2013). We utilised two parasite strains: an isolate from Cambodia (CAM3.II) with a mutated Kelch-13 conferring resistance, and a genetically modified revertant of this strain (CAM3.IIREV) conferring drug susceptibility (Straimer et al, 2015). Artemisinin resistance is difficult to detect in vitro using regular dose–response assays, and indeed we detected no difference between the artemisinin-resistant Kelch13 mutant C580Y (CAM3.II) and artemisinin susceptible Kelch13 revertant (CAM3.IIREV) parasites, nor the susceptible 3D7 strain, when treated with free artesunate or C-17-ART (Fig. EV2). The ring stage survival assay involves a short treatment window against newly-invaded ring stage parasites, followed by removal of drug pressure to mimic the clearance of the drug from plasma.

As reported (Witkowski et al, 2013; Straimer et al, 2015), Giemsa-stained smears of the resistant CAM3.II strain treated with artesunate or ART-link showed a combination of morphologically normal parasites (green arrows) and morphologically aberrant pyknotic or vacuolated forms, indicating loss of viability (orange arrows) (Fig. 4B). In contrast, upon treatment with C-17-ART, only aberrant, non-viable parasites were visible. Giemsa smears of the susceptible CAM3.IIREV strain confirmed that only morphologically aberrant parasites were present when treated with artesunate, C-17-ART, or ART-link.

We also used DNA replication to measure parasite survival and growth, revealing that a large proportion (~30%) of the resistant parasites survived 700 nM artesunate treatment (Fig. 4C). In marked contrast, complete growth inhibition was observed with the steroid-artesunate conjugate C-17-ART ($p = 0.002$ vs. artesunate; $p < 0.001$ for each vs. DMSO solvent control). In the susceptible control strain, all artesunate-containing compounds displayed complete growth inhibition ($p < 0.001$ for all compared to DMSO solvent control), whereas the steroid-linker compound C-17-link did not significantly impact the growth of either strain ($p = 0.08$ and $p = 0.22$ for resistant and susceptible strains, respectively).

Together, these data show that the artesunate conjugate was more effective against resistant parasites than the free drug, providing evidence that the steroid conjugation strategy may help to overcome drug resistance.

## Novel steroid-peroxide compound which mimics cholesterol inhibits *P. falciparum* growth

While improving existing antimalarial drugs has many conceivable clinical benefits, there is also potential for the design and synthesis of novel small molecules with antimalarial activity that exploit cholesterol uptake. Drugs such as primaquine and artesunate contain functional groups, responsible for antimalarial activity, with the rest of the molecule providing structure, stability, and favourable chemical properties to assist the functional group in reaching the target site. Since the steroid backbone contributes to the latter roles, we hypothesised that our conjugation strategy could be used to deliver functional groups as chemical warheads instead of whole drug molecules.

Synthetic reactive groups such as trioxolanes and trioxanes (like in artesunate) have been shown to effectively kill intra-erythrocytic *P. falciparum* parasites (Vennerstrom et al, 2004; Kaiser et al, 2007). Here, we took inspiration from this and coupled the steroid DHEA to a side chain containing a dialkyl peroxide (compound C-17-perox) in five steps with a 28% overall yield (see APPENDIX) in a way that more closely resembles the native structure of cholesterol (Fig. 5A). For comparison, we used di-*tert*-butyl peroxide to establish any general peroxide-dependent effects.

We tested these compounds and steroid controls for inhibitory effects on *P. falciparum* growth (Fig. 5B). We found that the steroid-bound peroxide C-17-perox inhibited parasite growth with an $IC_{50}$ of 0.48 µM. Neither cholesterol nor di-*tert*-butyl peroxide inhibited parasite growth within the range of concentrations tested ($IC_{50} > 400$ µM and 25 µM respectively), and DHEA exhibited an $IC_{50}$ of 150 µM. This suggests that a peroxide group capable of generating reactive oxygen species can be delivered to the intracellular parasite by incorporation into steroids. We also tested cytotoxicity against HEK293 cells, finding a ~60-fold difference in the C-17-perox $IC_{50}$ compared to *P. falciparum* (31.3 µM vs 0.5 µM), indicating good selectivity (Fig. 5C).

## Discussion

In this study, we investigated the uptake of cholesterol into *P. falciparum* parasites and devised a strategy that uses this pathway to efficiently deliver antimalarial drugs. This work represents a previously unrecognized way of exploiting apicomplexan cholesterol auxotrophy as an Achilles heel for targeting intracellular parasites. Instead of disrupting the movement or metabolism of molecules, we have instead sought to hijack an essential uptake pathway as a system for effectively delivering molecules with antiparasitic properties that work against a range of targets, and holds promise for a range of pathogens and diseases that remain

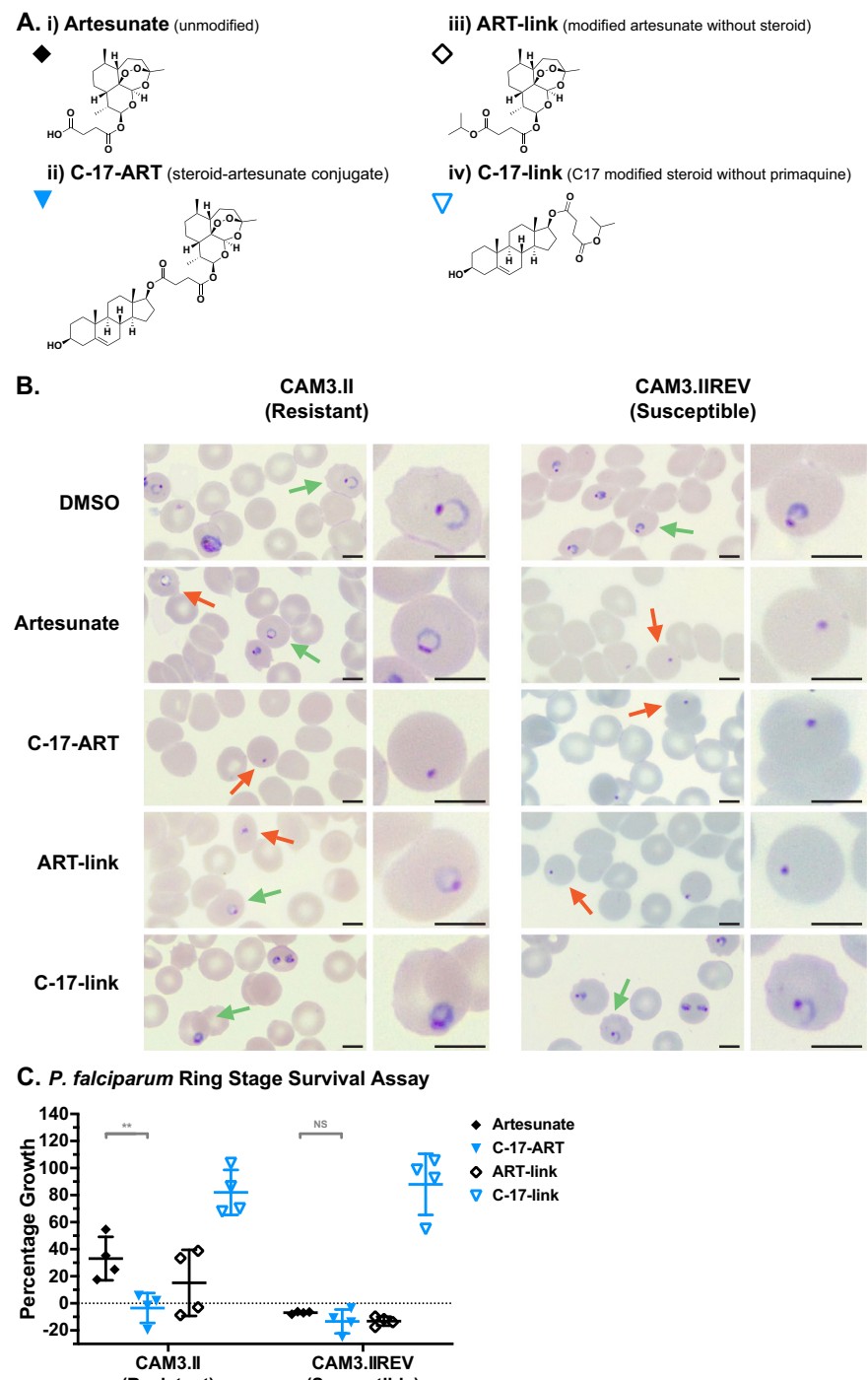

**Figure 4.  Steroid coupling of artesunate increases its potency against resistant *P. falciparum*.**

(**A**) Chemical structures of (i) artesunate; (ii) steroid with artesunate conjugated from C-17 (C-17-ART), (iii) artesunate with a linker (ART-link), and (iv) steroid molecule with a linker at C-17 (C-17-link). (**B**) Giemsa smears (overview and inset) of cultures from ring stage survival assays (6 h 700 nM drug treatment of newly invaded rings (0–3 h post invasion) followed by 66 h uninhibited growth) of asexual *P. falciparum* parasites resistant (CAM3.II) and susceptible (CAM3.IIREV) to artesunate. Green arrows = morphologically normal (viable) parasites; orange arrows = morphologically aberrant (vacuolated or pyknotic, non-viable) parasites. Scale bar = 10 μm. (**C**) Quantification of ring stage survival using DNA replication relative to the solvent control. Shown are mean values (± S.D.). NS, not significant; **$p < 0.01$ (ANOVA). $n = 2$ independent experiments. Source data are available online for this figure.

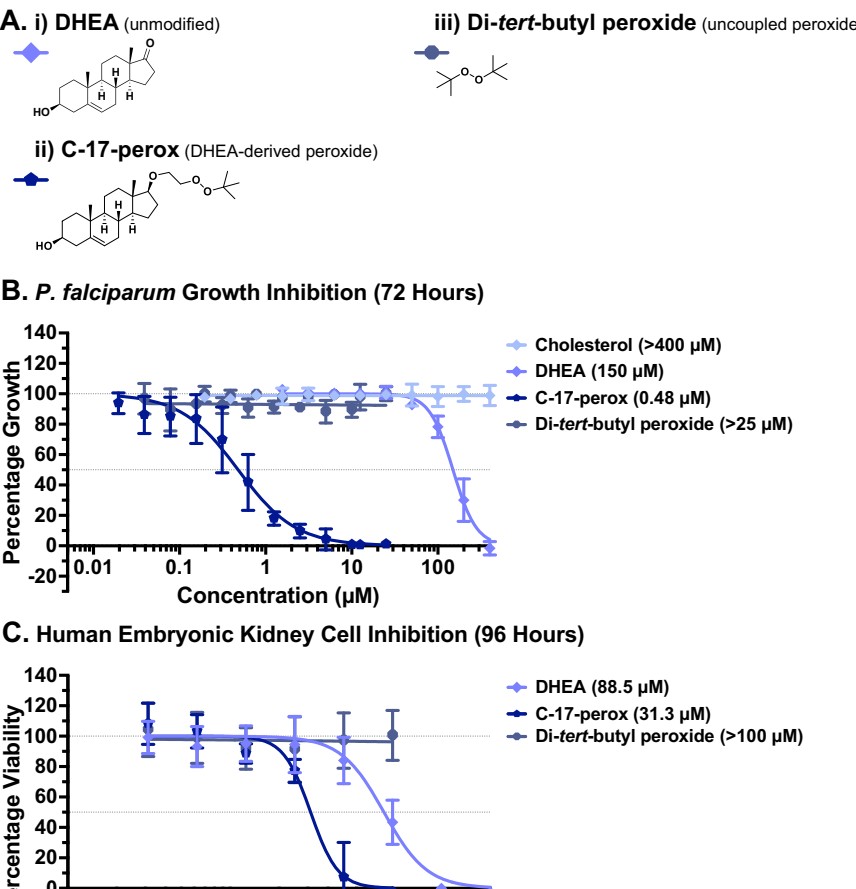

**Figure 5. Steroid-coupled peroxides inhibit *P. falciparum* growth.**

(A) Structure of (i) DHEA, (ii) DHEA-derived peroxide (C-17-perox), and (iii) di-*tert*-butyl peroxide. (B) Dose–response curve (72 h) of compounds against asexual *P. falciparum* growth. Shown are mean values (± S.D.). IC$_{50}$ is in brackets. $n = 3$ independent experiments. (C) Dose–response viability curve (96 h) against human embryonic kidney (HEK293) cells incubated with DHEA, C-17-perox, and di-*tert*-butyl peroxide. Shown are mean values (± S.D.). IC$_{50}$ is in brackets. $n = 4$ independent experiments. Source data are available online for this figure.

difficult to treat, including *Plasmodium*, *Toxoplasma gondii*, *Trypanosoma brucei* (sleeping sickness), and *Mycobacterium tuberculosis* (tuberculosis) (Maier and van Ooij, 2022; Bottova et al, 2009; Coppens and Courtoy, 2000; Miner et al, 2009).

We found that coupling the antimalarial drug primaquine to a cholesterol-mimicking steroid resulted in a compound that improves efficacy against *Plasmodium* parasites at three different lifecycle stages—asexual intraerythrocytic parasites, sexual gametocyte stages, and liver stages – while also showing lower cytotoxic effects to human cell lines. Strategies that are effective against multiple stages of parasites are highly desirable in the fight against these diseases (Burrows et al, 2017; Forte et al, 2021). We hypothesise that the shared reliance on host cholesterol unites these stages, and therefore the potential applications of this conjugation strategy are very broad. Since many antimalarial drugs in use are most effective against asexual blood stages, this strategy could make it possible to effectively target these other stages with the same drug. This could reduce the risk of transmission to a new host by killing gametocytes or avoid the symptomatic phase of

infection entirely by killing the parasites in the liver before they have a chance to progress to the blood stages. It will be interesting to explore the potential for the steroid-linked primaquine conjugates in the therapy of other *Plasmodium* species and whether the conjugation approach might for example boost the anti-relapse activity against dormant liver stages (hypnozoites) of *Plasmodium vivax*.

The steroid-coupling strategy also has the potential to improve the action of existing antimalarial compounds in the blood stages. The use of primaquine as an antimalarial treatment or preventative is limited by its slow action and high toxicity (Ashley et al, 2014; Chen et al, 2015; Vale et al, 2009). These properties could potentially be mitigated with better delivery to the parasite, so this drug was a good candidate for initial proof-of-principal studies. Steroid-coupled primaquine inhibits parasites more effectively than the free drug and therefore could allow for lower doses and address safety concerns associated with primaquine toxicity.

We also noted that the shape of the drug inhibition curve was much steeper with the C-17-PQ conjugate compared to primaquine

alone. Having a larger concentration range where some parasites survive drug exposure (exhibited by a flatter dose–response curve) indicates a potential higher risk of the emergence of drug-resistant parasites. Improved delivery of drug to the parasite results in the conjugate acting more efficiently than primaquine alone. Increased killing speed could provide an advantage for fast parasite clearance and decreases the chance of resistant parasites emerging.

The activity of primaquine is believed to be mediated by enzymatic and non-enzymatic biotransformations resulting in intermediates which react with molecular oxygen to produce hydrogen peroxide ($H_2O_2$), causing oxidative damage to the parasites (Camarda et al, 2019). Since enhanced activities of the conjugates compared to free primaquine are observed in the presence of different host cells (RBCs, hepatic cells, fibroblasts), providing different environments for the biotransformation (Fasinu et al, 2019; Wang et al, 2021), the obtained data also suggests that steroid conjugation may not interfere with the production of reactive oxygen species in vivo and hence primaquine action.

We found that coupling the antimalarial drug artesunate to a steroid improved its efficacy against parasites resistant to this drug, providing evidence that a delivery system may help to overcome drug resistance. The improved action of C-17-ART against the resistant parasites is potentially due to increased compound uptake within the short treatment window, and/or increased compound retention within the cell after removal from the extracellular environment. Since artesunate has a very short half-life in blood (0.5–1.5 h, (Morris et al, 2011)), either of these changes would pose an advantage for greater treatment efficacy against susceptible and resistant parasites. Clearly, further investigation into pharmacokinetic, pharmacodynamic and cytotoxic properties of the C-17 steroid-artesunate conjugate by employing comprehensive ADMET (administration, distribution, metabolism, excretion, toxicity) and similar studies is warranted. Swift preclinical development and exploratory phase 1 and 2 clinical trials are urgently needed for improved artemisinin-based combination therapies, and the C-17 steroid-artesunate conjugate holds promise to improve efficacy both against susceptible and resistant parasites. Steroid-coupling of other antimalarial drugs might reveal other applications of this approach and may be particularly useful in situations where drug resistance is related to compound uptake or retention. Efficient drug delivery could also address potential shortcomings of targeted drug discovery strategies: potent and specific inhibitors against parasite targets often fail in further experiments due to insufficient uptake by the parasite (Spangenberg et al, 2013; Burrows et al, 2017; Forte et al, 2021). Evidence-based strategies to deliver drugs to parasites will become more important as the field moves away from whole cell drug screens to more target-based approaches.

We have also presented evidence that this delivery system is broadly applicable across not only multiple lifecycle stages but across other parasite species, exemplified by *T. gondii*, where the steroid-primaquine conjugate showed a 3-fold lower $IC_{50}$ than primaquine. There are many drugs or drug-like compounds that are effective against *Plasmodium*, but not *Toxoplasma* or other closely related parasites, despite a high number of closely conserved genes (Boyom et al, 2014; Subramanian et al, 2018; Radke et al, 2018). In some cases, this is due to lower permeability and drug uptake, preventing these compounds from accessing their target site (Baumeister et al, 2011; Nair et al, 2011). The steroid-based drug delivery system may be especially useful in these cases since

antimalarial drugs could be repurposed for toxoplasmosis and related diseases, and vice versa. In addition, many other (non-apicomplexan) parasites and pathogens also rely on the uptake of host cholesterol, and so the delivery system might also be relevant to these pathogens (Bansal et al, 2005; O'Neal et al, 2020). While there will certainly be restrictions on suitable drugs and target organisms, the potential for broad application is intriguing.

We also describe a strategy of directly modifying steroids into drug-like compounds by attaching peroxides directly onto the steroid core. These compounds likely take advantage of the parasites' increased susceptibility to oxidative damage by delivering a molecule that may produce reactive oxygen species. These compounds have a lower $IC_{50}$ than primaquine against asexual *P. falciparum*, highlighting their potential as a therapeutic compound, although further chemical modifications and mechanistic studies of this proof-of-concept would likely be needed before deployment as an antimalarial drug.

Our study also provided insights into the structural requirement for parasite cholesterol uptake: fluorescent cholesterol conjugate uptake was higher when the fluorophore was conjugated from the C-17-linked side chain of cholesterol, leaving the C-3 hydroxy group unmodified. This difference in uptake based on conjugation sites was consistent in the activity of the steroid-primaquine conjugates. The C-3 hydroxy group has previously been implicated as important in the recognition of cholesterol by human transport enzymes (Kwon et al, 2009), as well as a wide range of cholesterol-binding proteins from various species (Bukiya and Dopico, 2017). Judging from our data, having an unmodified C-3 hydroxy group seems to also be important for the uptake or trafficking to the parasite. This requirement might also explain why attempts to conjugate drugs to the C-3 group of cholesterol were not successful (Morake et al, 2018). The importance of the free C-3 hydroxy group is further corroborated by a study finding that the effects of cholesterol depletion on *Plasmodium* could not be recovered with reconstitution using epicholesterol, which differs from cholesterol in the stereochemistry at the C-3 site (Ahiya et al, 2022). Together, this suggests that the spatial configuration of the C-3 hydroxy group is an important feature of cholesterol for parasite development, thus conjugating compounds on or after the C-17 of the cholesterol sidechain is a more suitable method for drug delivery.

In conclusion, we present a strategy of conjugating molecules to a steroid backbone in order to exploit cholesterol uptake by *Plasmodium* for drug delivery. This strategy improved the efficacy of the current antimalarial drugs primaquine and artesunate and can be used to deliver other drug-like molecules to the parasite. The strategy could be broadly applicable to a wide variety of other parasites and pathogens that rely on cholesterol scavenging from the host (Coppens, 2013; Ehrenman et al, 2013; Mondal et al, 2016). If confirmed in clinical studies, this strategy has the potential to substantially improve drug therapies for treating livestock diseases and human diseases such as malaria.

# Methods

## Ethics statement

All relevant aspects of this study were approved by the Australian National University's Human Ethics Committee, procedure

HEC2017/351, or the Berlin Ethics Committee (Landesamt für Gesundheit und Soziales Berlin), permit G0294/15, in strict adherence to German and European Union animal protection laws. Human red blood cell and serum were kindly provided by the Australian Red Cross Blood Service ("Lifeblood"). Donor consent was obtained as part of the donation process.

## Chemical synthesis

The steroid drug conjugates and related control compounds were prepared by chemical synthesis, with compound identity and purity (>95%) confirmed for all materials prior to biological evaluation. Reaction schemes, detailed experimental procedures, and characterisation data for synthesised materials are provided in the APPENDIX.

## Cell culture

Human hepatoma cells (Huh7), human foreskin fibroblasts (HFF), human embryonic kidney cells (HEK293), *P. falciparum* parasites (in human RBCs), and *T. gondii* parasites (in HFF cells) were maintained under routine culture conditions as detailed in the APPENDIX. All experiments used *P. falciparum* 3D7 wild-type parasites unless otherwise specified. Gametocyte formation was induced as described (Ridgway et al, 2020). *Plasmodium berghei* ANKA Bergreen (Kooij et al, 2012) or GFP-Luc (RMgm-29) (Janse et al, 2006), respectively expressing green fluorescent protein (GFP) or a GFP-Luciferase fusion protein, were maintained under routine conditions in SWR/J *Mus musculus* mice and *Anopheles stephensi* mosquitos.

## Uptake of fluorescent cholesterol analogues

In brief, ring-stage parasites at 4% parasitemia and 2% haematocrit were incubated with 2 µM fluorescent cholesterol analogue (dehydroergosterol, 24-BODIPY-cholesterol, 3-undecanoate-BODIPY-cholesterol, 22-NBD-cholesterol, or hexanoyl-NBD-cholesterol), 6 µM NBD without cholesterol (p-methoxybenzylamine-NBD), or solvent controls in cholesterol-depleted media for 24 h at 37 °C under standard culturing conditions. Cells were washed twice in PBS with 10 mM D-Glucose (PBS-G), and then resuspended in 5 µg/mL Hoechst 33342 (Thermo Fisher 14533) in PBS-G and incubated for 20 min at 37 °C. Cells stained with dehydroergosterol were instead incubated with 500 nM Mitotracker™ Deep Red FM (Thermo Fisher M22426) for 15 min and then washed twice in PBS-G. Samples were imaged with deconvolution fluorescence microscopy (×1000 magnification) or read on a flow cytometer, using Hoechst 33342 fluorescence to differentiate uRBCs (negative) and iRBCs (positive). For each fluorescent cholesterol analogue, images were collected under the same exposure conditions (without binning) and converted to TIFF files under the same brightness and contrast settings. Individual cells were cropped from larger images with Fiji ImageJ. No other manipulations were performed. Flow cytometry data was normalised by subtracting background (unstained) fluorescence and by setting the fluorescence of uRBCs to 1. Detailed methods are provided in the APPENDIX.

## Growth inhibition and viability assays

Growth inhibition or cell viability were assessed by incubation with a serial dilution of test compounds, solvent controls (DMSO, ethanol, ethyl acetate), or culture media alone in 96-well plates, with detection of fluorescence, luminescence, or absorbance in microplate readers after 24–96 h unless otherwise specified.

*P. falciparum* asexual parasite assays commenced with ring-stage parasites at 1% parasitaemia and 1% haematocrit. 200 nM chloroquine (Sigma C6628) or artesunate (Sigma A3731) was used as a no-growth control. DNA replication (indicating growth) was measured at the specified time using SYBR Safe DNA Gel Stain (Invitrogen 33102) and a fluorescence microplate reader (Smilkstein et al, 2004; Spry et al, 2013).

Gametocyte assays commenced seven days (stage III–IV) or ten days (stage V) post-invasion. 100 mM artemisinin (Sigma 361593) was used as no-viability control. Mitochondrial membrane potential (indicating viability) was measured using flow cytometry after staining with 500 nM MitoTracker™ Deep Red FM and 5 µg/ml Hoechst 33342 to detect DNA (Ridgway et al, 2020).

Huh7 cells were seeded at a density of 10,000 cells per well of a 96-well plate, or 20,000 cells per collagen-coated coverslip. After 24 h, test compounds were added the wells were inoculated with *P. berghei* liver stage sporozoites dissected from infected *A. stephensi* salivary glands with 5000–10,000 sporozoites per well or 10,000–20,000 sporozoites per coverslip. Uninfected wells served as a control for background luminescence. Luminescence signal was measured 48 h post-invasion by addition of 1:1 ONE-Glo™ substrate (Promega) with a luminescence microplate reader. Coverslips were fixed, stained, and imaged, with all parasites on each coverslip imaged and delineated as described (Petersen et al, 2017) using Fiji ImageJ (Schindelin et al, 2012). Data are presented as a percentage of the size or parasite number of the solvent control to normalize for differences between experiments. For cell images presented in the figure, images were collected under optimal exposure settings to ensure features were visible. Individual cells were cropped from larger images with Fiji ImageJ, and the brightness was adjusted while ensuring that no fluorescence was removed. No other manipulations were performed.

*T. gondii* tachyzoite parasite assays commenced by adding 2000 RH strain tachyzoites expressing the fluorescent protein tdTomato to each well of a 96-well plate containing confluent HFF cells (Rajendran et al, 2017; Hayward et al, 2023). Fluorescence was measured daily with a fluorescence microplate reader for one week, and the background fluorescence from the time 0 reading was subtracted from all other measurements. The 72-h timepoint was chosen for further analysis because parasite growth was in the mid-logarithmic stage (Fig. EV3).

Huh7 hepatoma viability assays were conducted in the same manner as the *P. berghei* assays, with metabolic capacity (indicating viability) measured with a fluorescence microplate reader after incubation with CellTiter-Blue® (Promega) for 3 h.

HFF cell growth inhibition assays commenced by adding 5000 cells to each well of a 96-well plate. 10 µg/mL cycloheximide was used as a no-growth control. DNA replication (indicating growth) was measured at the specified time using SYBR Safe DNA Gel Stain (Invitrogen 33102) and a fluorescence microplate reader.

HEK293 cell viability assays commenced with 5000 cells per well. 10 µg/mL cycloheximide was used as a no-growth control. Metabolic capacity (indicating viability) was measured in an absorbance microplate reader after incubation with 0.45 mg/mL methylthiazolyldiphenyl-tetrazolium bromide (MTT; Sigma M2128) for 2 h followed by crystal solubilisation in SDS.

**The paper explained**

**Problem**
Resistance to antimalarial drugs is a formidable barrier to the goal of eradicating malaria, a devastating disease caused by *Plasmodium* parasites. These parasites rely on the uptake of host nutrients to survive, including the lipid cholesterol.

**Results**
We first established that cholesterol molecules with bulky fluorescent moieties could be delivered to intracellular parasites inside human red blood cells. In place of the fluorophore, we conjugated existing antimalarial drugs, primaquine and artesunate, to steroids that mimic the structure of cholesterol. Conjugated primaquine showed increased in vitro inhibitory effects compared to unconjugated primaquine against parasites at multiple lifecycle stages which occur in the human host, showing lower toxicity to human cells. Conjugated artesunate was able to overcome drug resistance in a parasite strain that displays clinically relevant resistance to unconjugated artesunate. The cholesterol mimic strategy can also be adapted to create new antimalarial compounds using synthetic reactive groups instead of existing drugs.

**Impact**
These findings provide a proof-of-concept for a drug delivery strategy that hijacks cholesterol uptake pathways to deliver antimalarial compounds to the parasite, increasing their efficacy and safety. This strategy could improve and revitalise existing antimalarial treatments, create novel drugs, and ultimately reduce the burden of malaria and other parasitic diseases.

### *Plasmodium falciparum* ring stage survival assay

Following tight synchronisation using sorbitol and Percoll® floatation, ring stage CAM3.II and CAM3.IIREV parasites (0–3 h post invasion) at 1% parasitaemia and 2% haematocrit were incubated with 700 nM of compounds or solvent-only controls for exactly 6 h, before removing the drug, thoroughly washing in RPMI with HEPES, and returning to culture conditions for a further 66 h. Some cells were treated with 700 nM artesunate for 72 h as a no-growth control. DNA replication (indicating growth) was measured at the specified time using SYBR Safe DNA Stain. Microscope slide smears of each condition were stained with 10% v/v Giemsa and examined under a light microscope (×1000 magnification); with 'viable' (morphologically normal) and 'non-viable' (morphologically aberrant; vacuolated or pyknotic) parasites distinguished according to standard protocol (Witkowski et al, 2013). Images were captured using a Leica ICC50 camera, with a resolution of 76 nm per pixel, and cropped from larger images using Fiji ImageJ.

### Growth inhibition data

Growth inhibition data were normalised by subtracting background fluorescence, luminescence, absorbance, or percentage of cells from a no-growth or no-viability control (high drug concentration, no-parasite control, or time = 0 measurement as specified), and expressed as a percentage of the growth or cell percentage in the negative control (culture media alone or solvent control) at the indicated timepoint (or to the highest measurement reached during the assay for Fig. EV3). Data were fitted with a four parameter

[inhibitor] vs response curve in Prism 9, and the concentration inhibiting 50% of growth ($IC_{50}$) was calculated by the model.

### Statistics

Where appropriate, quantification was represented by the mean ± standard deviation. Except where otherwise specified, data were analysed in Prism 9 using one-way or two-way ANOVA with corrections for the false discovery rate using the two-stage setup method of Benjamini et al (Benjamini et al, 2006). Due to the nature of the experiments, no randomisation or blinding took place.

### Data availability

This study includes no data deposited in external repositories.

The source data of this paper are collected in the following database record: biostudies:S-SCDT-10_1038-S44321-024-00087-1.

### Peer review information

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

## Acknowledgements

The authors wish to thank Daniela Číhalová, Melanie Rug, Daryl Webb, Michael Devoy, Harpreet Vohra, Victor Makota, and Imam Fathoni for advice on methodological aspects; Gregory Gunawan and Hayden Bell for assistance with chemical synthesis; David Fidock and Leann Tilley for providing the CAM3.II and CAM3.IIREV cell lines; Andreas Herrmann and Peter Müller for helpful discussions; the Australian Red Cross for providing human RBCs and serum. We would like to acknowledge the following funding bodies for support for this project: Alliance Berlin Canberra "Crossing Boundaries: Molecular Interactions in Malaria", co-funded by a grant from the Deutsche Forschungsgemeinschaft (DFG) for the

International Research Training Group (IRTG) 2290 and the Australian National University (GGvD, AI, KM, MM, AGM); Deutsche Forschungsgemeinschaft (DFG) Research Traing Group GRK 2046 (AI); Australian Research Council grant DP180103212 (AGM); National Health and Medical Research Council of Australia grant GNT1182369 (AGM, GGvD); Australian Government Research Training Program Scholarships (MF, BC, PAY, KSL).

## Author contributions

**Merryn Fraser**: Formal analysis; Investigation; Visualization; Methodology; Writing—original draft; Writing—review and editing. **Blake Curtis**: Formal analysis; Investigation; Visualization; Methodology; Writing—review and editing. **Patrick Phillips**: Investigation; Writing—review and editing. **Patrick A Yates**: Investigation; Methodology; Writing—review and editing. **Kwong Sum Lam**: Investigation; Visualization; Writing—review and editing. **Otto Netzel**: Investigation; Writing—review and editing. **Giel G van Dooren**: Resources; Methodology; Writing—review and editing. **Alyssa Ingmundson**: Resources; Supervision; Methodology; Writing—review and editing. **Kai Matuschewski**: Methodology; Writing—review and editing. **Malcolm D McLeod**: Conceptualization; Resources; Supervision; Methodology; Project administration; Writing—review and editing. **Alexander G Maier**: Conceptualization; Resources; Formal analysis; Supervision; Funding acquisition; Visualization; Methodology; Writing—original draft; Project administration; Writing—review and editing.

Source data underlying figure panels in this paper may have individual authorship assigned. Where available, figure panel/source data authorship is listed in the following database record: biostudies:S-SCDT-10_1038-S44321-024-00087-1.

## Disclosure and competing interests statement

MF, BC, PAY, MM and AGM are named inventors on a patent application (PCT/AU2023/050041) by the Australian National University that covers the technology described.

# Expanded View Figures

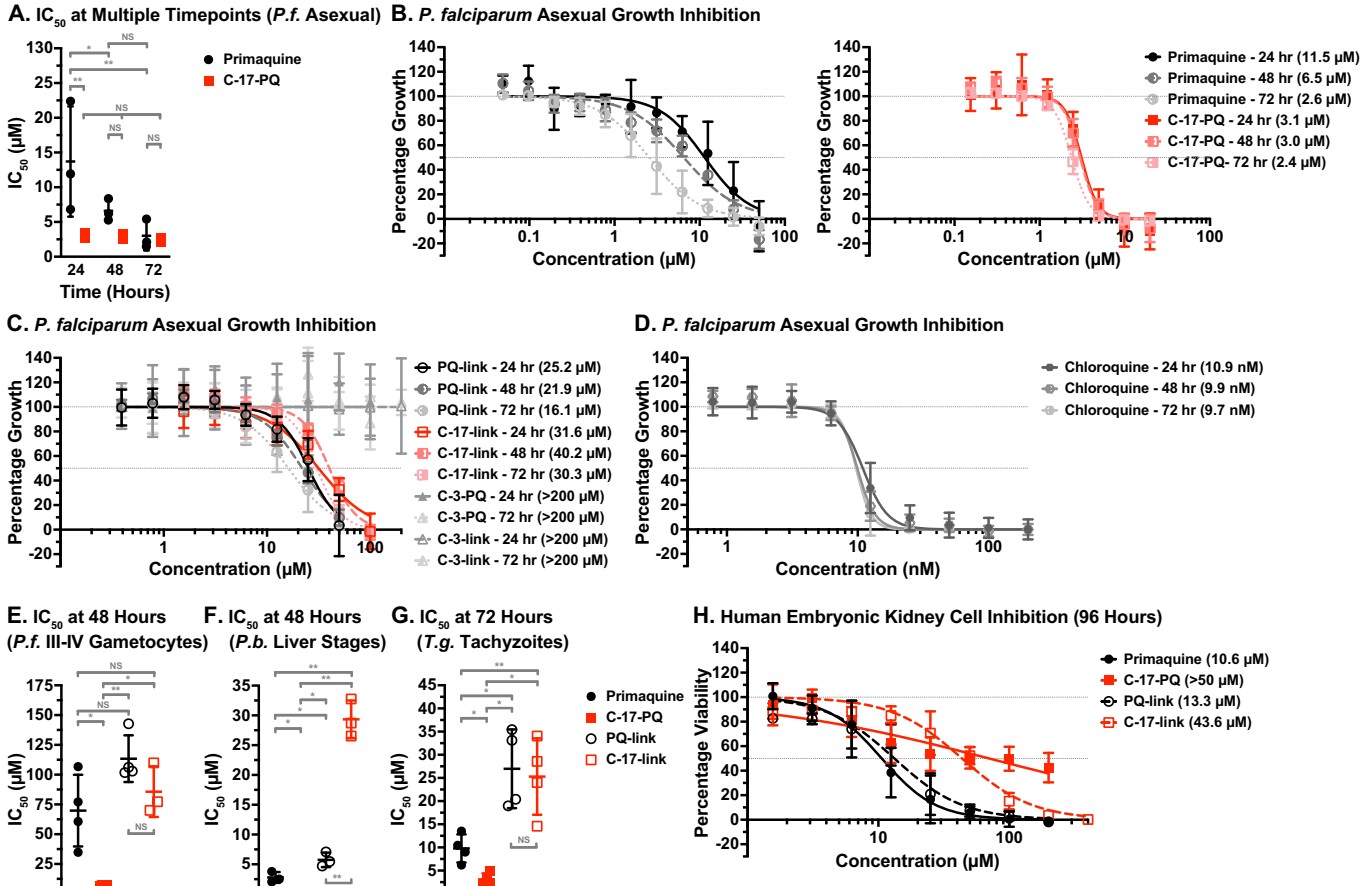

**Figure EV1. Coupling of primaquine to a steroid increases its potency against apicomplexan parasites.**

(A) Comparison of 50% inhibitory concentrations ($IC_{50}$) values against asexual *P. falciparum* from Fig. 2D. Individual data points are the $IC_{50}$ values from independent experiments; centre bar is the mean value (± S.D.) of these data points. (B–D) Dose–response curves of (B) primaquine, C-17-PQ, (C) PQ-link, C-17-link, C-3-PQ, C-3-link, and (D) chloroquine against asexual *P. falciparum* growth at 24, 48 and 72 h. Shown are mean values (± S.D.). $IC_{50}$ values are in brackets. $n = 3$ independent experiments. (E) Comparison of $IC_{50}$ values against sexual *P. falciparum* gametocytes from Fig. 2C. Individual data points are the $IC_{50}$ values from independent experiments; centre bar is the mean value (± S.D.) of these data points. $n = 4$ independent experiments. (F) Comparison of $IC_{50}$ values against *P. berghei* liver stages from Fig. 3A. Individual data points are the $IC_{50}$ values from independent experiments; centre bar is the mean value (± S.D.) of these data points. $n = 3$ independent experiments. (G) Comparison of $IC_{50}$ values against *Toxoplasma gondii* from Fig. 3F. Individual data points are the $IC_{50}$ values from independent experiments; centre bar is the mean value (± S.D.) of these data points. $n = 4$ independent experiments. (H) Dose–response assay (96 h) against human embryonic kidney (HEK293) cell viability incubated with primaquine, C-17-PQ, PQ-link, or C-17-link. Shown are mean values (± S.D.). $IC_{50}$ is in brackets. $n = 3$ independent experiments. NS, not significant; *$p < 0.05$; **$p < 0.01$ (ANOVA). Source data are available online for this figure.

**P. falciparum Asexual Growth Inhibition (72 Hours)**

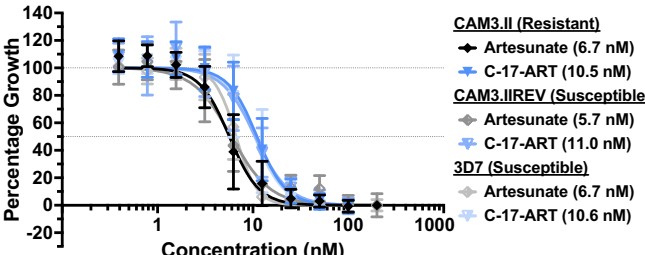

Figure EV2. **Dose–response assay (72 h) of artesunate and C-17-ART against resistant (CAM3.II) and susceptible (CAM3.II REV and 3D7) asexual *P. falciparum* growth (starting at ring stages).**

Shown are mean values (± S.D.). 50% inhibitory concentration ($IC_{50}$) is in brackets. $n = 4$ independent experiments. Source data are available online for this figure.

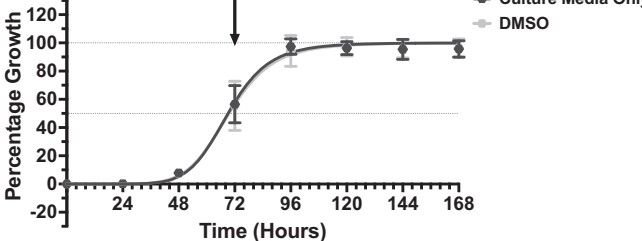

**_T. gondii_ Growth Over Time**

**Figure EV3.   Measurement of _Toxoplasma gondii_ tachyzoite growth in supplemented media, or supplemented media with 0.1% (v/v) DMSO (solvent control for drugs) over one week.**

Parasites were in the mid-logarithmic phase of growth at 72 h post seeding (arrow), hence this timepoint was used for the experiment outlined in Fig. 3F. Shown are mean values (± S.D.) $n = 4$ independent experiments. Source data are available online for this figure.

