## [Peer Review File · EMBO Molecular Medicine]

Harnessing cholesterol uptake of malaria parasites for therapeutic applications

Merryn Fraser, Blake Curtis, Patrick Phillips, Patrick Yates, Kwong Lam, Otto Netzel, Giel van Dooren, Alyssa Ingmundson, Kai Matuschewski, Malcolm McLeod, and Alexander Maier

Corresponding authors: Alexander Maier (alex.maier@anu.edu.au) , Malcolm McLeod (malcolm.mcleod@anu.edu.au)

Review Timeline:

Submission Date:	18th Oct 23
Editorial Decision:	21st Nov 23
Revision Received:	30th Mar 24
Editorial Decision:	10th Apr 24
Revision Received:	14th May 24
Accepted:	24th May 24

Editor: Zeljko Durdevic

Transaction Report:

21st Nov 2023

Dear Prof. Maier,

Thank you for the submission of your manuscript to EMBO Molecular Medicine, and please accept my apologies for the delay in getting back to you, which is due to the fact that one referee needed more time to complete his/her review. We have now received feedback from the two reviewers who agreed to evaluate your manuscript. Both referees recognize potential interest of the study but also raise serious criticism that should be addressed in a major revision. We agree with the referee #1 that in vivo experiments would significantly strengthen the findings and add much value to the study and would therefore encourage you to perform an in vivo experiment using P. berghei model of infection as suggested by the referee #1. If you would like to discuss further the points raised by the referees, I am available to do so via email or video. Let me know if you are interested in this option.

We would welcome the submission of a revised version within three months for further consideration. Please let us know if you require longer to complete the revision.

Please use this link to login to the manuscript system and submit your revision: <https://embomolmed.msubmit.net/cgi-bin/main.plex>

I look forward to receiving your revised manuscript.

Yours sincerely,

Zeljko Durdevic

We require:

- 1) A .docx formatted version of the manuscript text (including legends for main figures, EV figures and tables). Please make sure that the changes are highlighted to be clearly visible.
- 2) Individual production quality figure files as .eps, .tif, .jpg (one file per figure). For guidance, download the 'Figure Guide PDF': (<https://www.embopress.org/page/journal/17574684/authorguide#figureformat>).
- 3) A .docx formatted letter INCLUDING the reviewers' reports and your detailed point-by-point responses to their comments. As part of the EMBO Press transparent editorial process, the point-by-point response is part of the Review Process File (RPF), which will be published alongside your paper.
- 4) A complete author checklist, which you can download from our author guidelines (<https://www.embopress.org/page/journal/17574684/authorguide#submissionofrevisions>). Please insert information in the checklist that is also reflected in the manuscript. The completed author checklist will also be part of the RPF.
- 5) Please note that all corresponding authors are required to supply an ORCID ID for their name upon submission of a revised

manuscript.

6) It is mandatory to include a 'Data Availability' section after the Materials and Methods. Before submitting your revision, primary datasets produced in this study need to be deposited in an appropriate public database, and the accession numbers and database listed under 'Data Availability'. Please remember to provide a reviewer password if the datasets are not yet public (see <https://www.embopress.org/page/journal/17574684/authorguide#dataavailability>).

13) Author contributions: You will be asked to provide CRediT (Contributor Role Taxonomy) terms in the submission system. These replace a narrative author contribution section in the manuscript.

14) A Conflict of Interest statement should be provided in the main text.

Please also suggest a striking image or visual abstract to illustrate your article as a PNG file 550 px wide x 300-800 px high.

***** Reviewer's comments *****

Referee #1 (Comments on Novelty/Model System for Author):

The authors' findings should be verified in an in vivo model of Plasmodium infection. I would suggest the use of P. berghei parasites and C57BL/6 and/or BALB/c mice.

Referee #1 (Remarks for Author):

Merryn and collaborators explore Plasmodium's cholesterol pathways to achieve the delivery of antiplasmodial drugs to intracellular parasites. To investigate this, the authors attached either primaquine or artesunate to steroids that mimic the structure of cholesterol and found that this improved the effects of those drugs against various stages of the malaria parasite's life cycle. Their results further suggest that modified steroid molecules display antiplasmodial activity. The article is well-written and the conclusions are largely supported by the data. However, I have a few concerns that should be addressed before the manuscript can be considered suitable for publication in EMBO Mol Med.

One of my main concerns regards the use of primaquine to demonstrate the proof of principle of the benefits of coupling antiplasmodial compounds to steroids to increase their efficiency against blood stage Plasmodium infection. As the authors know, primaquine is not commonly employed to treat malaria, as it is not particularly active against blood stage Plasmodium parasites. Instead, it is active against gametocytes and liver stage parasites, which is why it is used for radical cure of infections by hypnozoite-forming Plasmodium species, such as P. vivax. Moreover, it is well known that primaquine needs to be metabolized in vivo in order to exert its antiplasmodial activity, as it is not primaquine, but rather its metabolites, that acts on infection. With all this in mind, it is unclear to me: (i) why the authors selected primaquine for this proof-of-principle demonstration for the blood stage of Plasmodium infection; and (ii) whether the authors have taken into consideration the requirement for primaquine metabolization when performing this study and, if so, why that was not duly acknowledged in the manuscript. Likewise, as the authors themselves acknowledge, "T. gondii is less susceptible to primaquine than Plasmodium, and this drug is not used for the treatment of toxoplasmosis". As such, it makes little sense to me to use primaquine as the drug of choice for a proof-of-principle demonstration that employs Toxoplasma as the target parasite.

My other concern regards the absence of in vivo data in the manuscript. While the authors employ adequate in vitro assays throughout their study, it is well known that bioavailability and other issues may compromise a compound's activity in vivo. This is particularly relevant if one bears in mind that some of the molecules generated and tested in this study are quite bulky, which may compromise their absorption in vivo. I do realize that testing inhibition of P. falciparum infection in vivo is challenging and requires the use of humanized mouse models, but the authors could have employed the P. berghei model of infection to assess their strategy in either blood or liver stage infection in rodents. I am aware that these are costly and time-consuming experiments and I can understand it if the authors chose not to perform them, but they would certainly add much value to their findings.

Other than these points, I am of the opinion that the paper contains useful and novel information and that these results are worthy of publication.

Referee #2 (Remarks for Author):

Reviewer comments:

Summary: In this manuscript, the authors propose an elegant and innovative way to boost the targeting of antimalarials such as primaquine and artesunate by hijacking the natural process by which malaria parasites harvest steroids (mainly cholesterol) from their host red blood cells. The authors approach is to chemically link cholesterol or other steroids to the primaquine and artesunate with the goal to improve the internalization of these antimalarial into the intracellular environment of Plasmodium. They demonstrate that when primaquine is coupled to steroids, its potency is enhanced by at least a factor 3 in in-vitro assays for the asexual, sexual and liver stages of either the human (asexual blood stage and gametocytes) or the rodent (liver stage) parasites while diminishing the cellular toxicity on 3 different human cell lines. Similarly, steroid-coupled primaquine inhibits the growth of Toxoplasma gondii, another apicomplexan parasite. Cholesterol coupled to artesunate abolishes the recrudescence of ring parasites containing mutations in the kelch13 gene when tested in the ring stage assay.

The article is well written, in a logical frame, and is easy to read. The study is conducted with the goal to verify working hypotheses and clear demonstrations are made that strongly support the conclusions. I would be very supportive of this manuscript to be published in EMBO Molecular Medicine provided that the authors answer the points raised below.

Major points:

1) In Figure 1, it seems that a control of staining with uncoupled (free) bodipy and NBD is necessary to assess the background (cholesterol independent) fluorescence in non-infected and infected red blood cells.

2) In figure 2, the authors test the modified primaquine linked to a linker attached to either one or the other extremities of the cholesterol molecules (C17 or C3) not only on asexual blood stage parasites but also on stage III-IV gametocytes. This is of great interest to assess the possibility to boost primaquine efficiency against these sexual stages and thus hoping to ultimately block transmission. However, testing the modified primaquine against stage V gametocytes which are the ones circulating in the blood of patients and responsible for malaria transmission when taken by the mosquito during a blood meal would have been even more relevant for assessing the potential boost of the transmission blocking activity of primaquine.

3) Lines 171 to 174: I agree with the authors that the most plausible explanation for the change of parasite killing rate by PQ is the coupling to a steroid that improves its absorption into the parasite however numerous PRR studies with antimalarial drug candidates have shown that the speed of kill is highly depending on the mode of action and the modification of the chemistry of the compounds will affect this PRR in case one sees a shift of the compounds from one target to another one. The authors should be prudent here and should consider that due to the modification of its scaffold, the steroid-coupled primaquine could be act by an alternative target/pathway. What is clear is that the steroids coupled to primaquine transform a slow acting drug into a fast acting one. A more sensitive and well validated assay would be to use the PRR assay for a better assessment of the speed and completeness of kill in vitro.

4) Line 175: as stated above, the entire section on gametocyte is highly valuable yet having studied the effect of coupling primaquine to steroids on stage V gametocytes which are the transmissible parasites would have been of additional value.

5) Line 253: In this section authors study whether steroid-artesunate conjugation overcomes drug resistance in Plasmodium ring stages. This is a very important aspect of their work and particularly timely while the world is facing the emergence and spread of partial artemisinin resistance which once coupled to partner drug frank resistance leads to clinical failures as it was recently observed in Cambodia. The experiments in this section are well conducted while not easy (the Ring Stage Assay is notoriously difficult and poorly reproducible if not mastered perfectly). A particularity of artemisinin resistance and the reason why it is called partial is that in contrast to frank drug resistance, when tested in vitro on Kelch13 parasites, artemisinin does not lead to any loss of potency (increased IC50) compared to wild type parasites. It would strengthen the manuscript if the authors could show that the steroid-linked artesunate is behaving similarly when tested in regular IC50 determination studies.

6) In the discussion (line 372), the authors wrote "Swift preclinical development and exploratory phase 1 and 2 clinical trials are urgently needed for improved artemisinin-based combination therapies, and the C-17 steroid-artesunate conjugate holds promise to improve efficacy both against susceptible and resistant parasites. This is of course something that all antimalarial community members would be eager to see however in the world of drug discovery and development it is well known that the best toll compounds is never ensured to become the best drug. Beyond PK/PD studies that the authors suggest as translational work for first in man proof of concept, they should mention toxicity studies. Compared to primaquine or artesunate, these compounds will be new and for instance lipophilicity (LogP) with steroid moiety will increase leading to an increase of the risk for hERG inhibition and thus potentially cardiovascular safety issues. Log P could easily be calculated based on the structure of the linked molecules. The authors should consider this point in their discussion. They should also discuss the possibility to rapidly assess the cell permeability parameter that will ensure good intestinal absorption (Caco2 assays for instance) by studying the apical to basal and basal to apical transit of the new compounds vs. artesunate and primaquine.

7) In the discussion, the authors should also discuss the tremendous potential for the steroid-linked primaquine to boost its anti-relapse activity against the dormant form (hypnozoites) of *P. vivax* in the liver. Some in vitro assays to assess this exquisite biology of *P. vivax* exist today.

Minor points:

1) In the legend of figure 1, a spelling of DIC (differential interference contrast) is needed for readers who are not microscopists.

2) Line 257, authors wrote that emergence and spread of mutations in the parasite protein Kelch-13 results in delayed clearance meaning that there is growing concern that treatment failures will develop. Authors should rephrase because today this is more than a growing concern. Few years ago, treatments with DHA-piperaquine were shown to lead to failure in Cambodia due to partial artemisinin resistance and resistance to piperaquine.

3) Line 274-275. The authors wrote "We utilised two parasite strains: an isolate from Cambodia (CAM3.II) with a mutated Kelch-13 conferring resistance, and a genetically modified revertant of this strain (CAM3.IIREV) conferring drug susceptibility (52)". It would be good to have an indication by the authors of what the mutation is. The most widespread in Southeast Asia so far is the Kelch13-C580Y. For non-expert readers, this information will be important.

4) Line 392 authors wrote: "...and so the delivery system might also relevant to these pathogens (66, 67)". This sentence should read: ...and so the delivery system might also be relevant to these pathogens (66, 67).

Response to Reviewers' comments**Fraser/Curtis et al.****“Harnessing cholesterol uptake of malaria parasites for therapeutic applications”****EMM-2023-18849**

We would like to thank the reviewers for their very constructive and useful comments!

We are glad that the reviewers have endorsed the relevance of our manuscript for the research field. We have addressed all the points raised as follows:

Referee #1:

The authors' findings should be verified in an in vivo model of Plasmodium infection. I would suggest the use of *P. berghei* parasites and C57BL/6 and/or BALB/c mice.

Authors' response:

We agree with the referee that this is the next milestone at which to aim. However, these are complex experiments, and any mouse work needs to be carefully planned. The formulation for lipophilic compounds like our conjugates will not be as straightforward as it is for water-soluble compounds, for example like pyrimethamine. Different modes of delivery (oral vs. i.v. vs. i.p.) will have to be explored; bioavailability, clearance rate, metabolism, pharmacodynamic, and most importantly, safety/clinical signs need to be tested prior to obtaining efficacy data. It also requires ethics clearance and appropriate

funding. All this is planned for follow-up work. Since there are also several avenues/lead compounds to follow, we think that this is out of the scope of the present study.

Our approach to preclinical drug development is to design and perform a comprehensive study that leads to conclusive results and informs clinical development. We prefer a cautious and sustained approach, and we present here a proof-of-concept study to show that metabolic dependency of an intracellular pathogen can be employed for targeted drug delivery. The proof-of-concept nature of our study has been highlighted in the text in line 35 (abstract), 126, 247, 394 and 450).

Our study clearly warrants an in-depth analysis of all lead compounds. Hence a systematic exploration will entail the analysis of three conjugated lead drugs with up to three life cycle stages and up to two parasite lines. This should be done in a systematic fashion and in partnership with industry to prioritize preclinical development.

Referee #1 (Remarks for Author):

Merryn and collaborators explore Plasmodium's cholesterol pathways to achieve the delivery of antiplasmodial drugs to intracellular parasites. To investigate this, the authors attached either primaquine or artesunate to steroids that mimic the structure of cholesterol and found that this improved the effects of those drugs against various stages of the malaria parasite's life cycle. Their results further suggest that modified steroid molecules display antiplasmodial activity.

The article is well-written and the conclusions are largely supported by the data.

However, I have a few concerns that should be addressed before the manuscript can

be considered suitable for publication in EMBO Mol Med.

One of my main concerns regards the use of primaquine to demonstrate the proof of principle of the benefits of coupling antiplasmodial compounds to steroids to increase their efficiency against blood stage Plasmodium infection. As the authors know, primaquine is not commonly employed to treat malaria, as it is not particularly active against blood stage Plasmodium parasites. Instead, it is active against gametocytes and liver stage parasites, which is why it is used for radical cure of infections by hypnozoite-forming Plasmodium species, such as *P. vivax*. Moreover, it is well known that primaquine needs to be metabolized in vivo in order to exert its antiplasmodial activity, as it is not primaquine, but rather its metabolites, that acts on infection. With all this in mind, it is unclear to me: (i) why the authors selected primaquine for this proof-of-principle demonstration for the blood stage of Plasmodium infection;

Authors' response:

Since we wanted a proof-of-principle drug with broad applications, primaquine was selected for several reasons:

- Primaquine is a clinically relevant and proven anti-malarial, which, as stated by the reviewer, is the primary choice for cure (hypnozoite treatment) of *P. vivax* malaria. The treatment is done for 14 consecutive days and, hence, far from perfect. We wanted to see whether we can further improve an already successful drug.

-

- Primaquine shows some activity against all life cycle stages in the human host stages *in vitro*; in particular gametocytes, and therefore can be used for transmission blocking, but further improvements are necessary.
-
- Primaquine displays significant off-target toxicity which could be mitigated by effective delivery.
- Primaquine might not require cleavage from the steroid portion for activity.
- Primaquine allows for an easy attachment from the terminal amine.
- Primaquine is commercially available as a low-cost starting material, a major criteria for anti-malarial drug development.
-

We have highlighted these considerations in the revised manuscript (line 126ff): “We selected primaquine as a proof-of-principle compound since this compound could be improved to increase efficacy, therapeutic index, and reduce off-target effects. Primaquine is also an attractive candidate due to its activity against all life cycle stages in the human host and in particular against transmissible gametocytes. It is unlikely to require cleavage from a carrier for it to be active and has a terminal amine, which facilitates the ease of conjugation.”

and (ii) whether the authors have taken into consideration the requirement for primaquine metabolization when performing this study and, if so, why that was not duly acknowledged in the manuscript.

Authors' response:

We have been considering the requirement for primaquine metabolization and had referred to this in the original manuscript (line 53; whole paragraph line 405ff.). We would like to stress again that we are not suggesting that primaquine is the drug of choice that should be put forward in the clinical development of our drug delivery system. It was chosen as a proof-of-principle drug that could highlight the capabilities and limitations of our system.

However, the metabolic conversion of primaquine is not restricted to the *in-vivo* activation of the liver. Fasinu et al. [Malar J. 2019; 18: 30.] have shown that primaquine activation can take place via direct oxidation in the red blood cells, independently of the cytochrome P450 system, and be converted into the active primaquine-5,6-orthoquinone. The reservations about primaquine neglect that we also have shown improvements for artesunate and newly developed steroid peroxides. Furthermore, in the liver stage assays, these compounds were co-cultured with hepatic cells, allowing the metabolic conversion to take place.

We have modified the text to clarify this point (line 53ff): “Primaquine is a pro-drug which is metabolised in the human body into activated forms, either by direct oxidation in RBCs (17) or by the cytochrome P450 system in the liver (18).”

Likewise, as the authors themselves acknowledge, "T. gondii is less susceptible to primaquine than Plasmodium, and this drug is not used for the treatment of toxoplasmosis". As such, it makes little sense to me to use primaquine as the drug of choice for a proof-of-principle demonstration that employs Toxoplasma as the

target parasite.

Authors' response:

Similarly to above, we are not proposing the primaquine-conjugate as a new clinical agent against *Toxoplasma* infection, but we wanted to showcase the capability of the drug delivery system (line 242ff). The use of the same drug conjugates makes a comparison between organisms possible and underlines the potential of the drug delivery system for drug repurposing.

My other concern regards the absence of in vivo data in the manuscript. While the authors employ adequate in vitro assays throughout their study, it is well known that bioavailability and other issues may compromise a compound's activity in vivo. This is particularly relevant if one bears in mind that some of the molecules generated and tested in this study are quite bulky, which may compromise their absorption in vivo. I do realize that testing inhibition of *P. falciparum* infection in vivo is challenging and requires the use of humanized mouse models, but the authors could have employed the *P. berghei* model of infection to assess their strategy in either blood or liver stage infection in rodents. I am aware that these are costly and time-consuming experiments and I can understand it if the authors chose not to perform them, but they would certainly add much value to their findings.

Authors' response:

As stated above, we agree, that this is the next milestone to aim for. However, these are complex experiments, and any mouse work needs to be carefully planned. The formulation for lipophilic compounds like our conjugates will not be as straightforward

as it is for water-soluble compounds like for example pyrimethamine. Different modes of delivery (oral vs. i.v. vs. i.p.) will have to be explored, bioavailability, clearance rate, metabolism, pharmacodynamic, and most importantly, safety/clinical signs need to be tested prior to obtaining efficacy data. It also requires ethics clearance and appropriate funding. All this is planned for follow-up work. Since there are also several avenues/lead compounds to follow, we think that this is out of the scope of the present study.

Our approach to preclinical drug development is to design and perform a comprehensive study that leads to conclusive results and informs clinical development. We prefer a cautious and sustained approach, and we present here a proof-of-concept study to show that metabolic dependency of an intracellular pathogen can be employed for targeted drug delivery. The proof-of-concept nature of our study has been highlighted in the text in line 35 (abstract), 126, 247, 394 and 450).

Our study clearly warrants an in-depth analysis of all lead compounds. Hence a systematic exploration will entail the analysis of three conjugated lead drugs with up to three life cycle stages and up to two parasite lines. This should be done in a systematic fashion and in partnership with industry to prioritize preclinical development.

Other than these points, I am of the opinion that the paper contains useful and novel information and that these results are worthy of publication.

We thank the reviewer for their thorough consideration of our manuscript.

Referee #2 (Remarks for Author):

Reviewer comments:

Summary: In this manuscript, the authors propose an elegant and innovative way to boost the targeting of antimalarials such as primaquine and artesunate by hijacking the natural process by which malaria parasites harvest steroids (mainly cholesterol) from their host red blood cells. The authors approach is to chemically link cholesterol or other steroids to the primaquine and artesunate with the goal to improve the internalization of these antimalarial into the intracellular environment of Plasmodium. They demonstrate that when primaquine is coupled to steroids, its potency is enhanced by at least a factor 3 in in-vitro assays for the asexual, sexual and liver stages of either the human (asexual blood stage and gametocytes) or the rodent (liver stage) parasites while diminishing the cellular toxicity on 3 different human cell lines. Similarly, steroid-coupled primaquine inhibits the growth of Toxoplasma gondii, another apicomplexan parasite. Cholesterol coupled to artesunate abolishes the recrudescence of ring parasites containing mutations in the kelch13 gene when tested in the ring stage assay.

The article is well written, in a logical frame, and is easy to read. The study is conducted with the goal to verify working hypotheses and clear demonstrations are made that strongly support the conclusions. I would be very supportive of this

manuscript to be published in EMBO Molecular Medicine provided that the authors answer the points raised below.

Major points:

1) In Figure 1, it seems that a control of staining with uncoupled (free) bodipy and NBD is necessary to assess the background (cholesterol independent) fluorescence in non-infected and infected red blood cells.

Authors' response:

We thank the reviewer for raising this important point. We have included the background control for non-cholesterol conjugated NBD, which is now shown as Figure 1G, displaying minimal background fluorescence.

2) In figure 2, the authors test the modified primaquine linked to a linker attached to either one or the other extremities of the cholesterol molecules (C17 or C3) not only on asexual blood stage parasites but also on stage III-IV gametocytes. This is of great interest to assess the possibility to boost primaquine efficiency against these sexual stages and thus hoping to ultimately block transmission. However, testing the modified primaquine against stage V gametocytes which are the ones circulating in the blood of patients and responsible for malaria transmission when taken by the mosquito during a blood meal would have been even more relevant for assessing the potential boost of the transmission blocking activity of primaquine.

Authors' response:

We have now included data showing the effect of the primaquine-conjugate on stage V gametocytes (Fig. 2D) and have referred to the observed activity in the text of the revised

manuscript (line 201ff): “Once gametocytes have reached stage V they are transmissible and can persist in the blood stream for several weeks. Hence, we also tested the activity of our conjugate against stage V gametocytes (Fig. 2D). C-17-PQ showed similar activity against this stage compared to the earlier stages, with an IC₅₀ of 6.8 μM. This IC₅₀ represents a > 2-fold reduction compared to primaquine alone after 48 hours of incubation. These data indicate that gametocyte inhibition was improved with the steroid conjugation strategy, with a more pronounced difference in earlier stages.”

3) Lines 171 to 174: I agree with the authors that the most plausible explanation for the change of parasite killing rate by PQ is the coupling to a steroid that improves its absorption into the parasite however numerous PRR studies with antimalarial drug candidates have shown that the speed of kill is highly depending on the mode of action and the modification of the chemistry of the compounds will affect this PRR in case one sees a shift of the compounds from one target to another one. The authors should be prudent here and should consider that due to the modification of its scaffold, the steroid-coupled primaquine could be act by an alternative target/pathway. What is clear is that the steroids coupled to primaquine transform a slow acting drug into a fast acting one. A more sensitive and well validated assay would be to use the PRR assay for a better assessment of the speed and completeness of kill in vitro.

Authors' response:

We agree with the reviewer that alternative targets need to be considered. However, since the primaquine mode of action is ill-defined and given the complexity of assay we are

unsure whether determining the parasite reduction ratio would provide a definitive answer on this matter. The use of control compounds (primaquine + linker, steroid + linker) make the improved absorption into the parasite the most likely reason for the enhanced killing speed.

4) Line 175: as stated above, the entire section on gametocyte is highly valuable yet having studied the effect of coupling primaquine to steroids on stage V gametocytes which are the transmissible parasites would have been of additional value.

Authors' response:

See above. The additional data is shown in Figure 2D.

5) Line 253: In this section authors study whether steroid-artesunate conjugation overcomes drug resistance in Plasmodium ring stages. This is a very important aspect of their work and particularly timely while the world is facing the emergence and spread of partial artemisinin resistance which once coupled to partner drug frank resistance leads to clinical failures as it was recently observed in Cambodia. The experiments in this section are well conducted while not easy (the Ring Stage Assay is notoriously difficult and poorly reproducible if not mastered perfectly). A particularity of artemisinin resistance and the reason why it is called partial is that in contrast to frank drug resistance, when tested in vitro on Kelch13 parasites, artemisinin does not lead to any loss of potency (increased IC50) compared to wild type parasites. It would strengthen the manuscript if the authors could show that the steroid-linked artesunate is behaving similarly when tested in regular IC50

determination studies.

Authors' response:

Upon the reviewer's recommendation, we have included conventional dose-response assays to determine the IC₅₀ of 3D7 wild type, Kelch13 mutant and Kelch13 revertant parasites. Indeed, no significant shift in IC₅₀ is observed in these assays when wild type and Kelch13 mutant are compared. We have included this data (now as Fig. EV2) and have discussed this in line 304ff in the revised manuscript: "We tested these compounds in a ring stage survival assay, which allows the detection of resistant parasites and is the standard method for investigating resistance to artemisinin and its derivatives (51). We utilised two parasite strains: an isolate from Cambodia (CAM3.II) with a mutated Kelch-13 conferring resistance, and a genetically modified revertant of this strain (CAM3.IIREV) conferring drug susceptibility (52). Artemisinin resistance is difficult to detect *in vitro* using regular dose response assays, and indeed we detected no difference between the artemisinin resistant Kelch13 mutant C580Y (CAM3.II) and artemisinin susceptible Kelch13 revertant (CAM3.IIREV) parasites, nor the susceptible 3D7 strain, when treated with free artesunate or C-17-ART (Fig. EV2). The ring stage survival assay involves a short treatment window against newly-invaded ring stage parasites, followed by removal of drug pressure to mimic the clearance of the drug from plasma."

6) In the discussion (line 372), the authors wrote "Swift preclinical development and exploratory phase 1 and 2 clinical trials are urgently needed for improved artemisinin-based combination therapies, and the C-17 steroid-artesunate conjugate

holds promise to improve efficacy both against susceptible and resistant parasites. This is of course something that all antimalarial community members would be eager to see however in the world of drug discovery and development it is well known that the best toll compounds is never ensured to become the best drug. Beyond PK/PD studies that the authors suggest as translational work for first in man proof of concept, they should mention toxicity studies. Compared to primaquine or artesunate, these compounds will be new and for instance lipophilicity (LogP) with steroid moiety will increase leading to an increase of the risk for hERG inhibition and thus potentially cardiovascular safety issues. Log P could easily be calculated based on the structure of the linked molecules. The authors should consider this point in their discussion. They should also discuss the possibility to rapidly assess the cell permeability parameter that will ensure good intestinal absorption (Caco2 assays for instance) by studying the apical to basal and basal to apical transit of the new compounds vs. artesunate and primaquine.

Authors' response:

We thank the reviewer for this suggestion and have now elaborated on these points in the discussion of the revised manuscript (line 419ff): “Clearly, further investigation into pharmacokinetic, pharmacodynamic and cytotoxic properties of the C-17 steroid-artesunate conjugate by employing comprehensive ADMET (administration, distribution, metabolism, excretion, toxicity) and similar studies is warranted. Swift preclinical development and exploratory phase 1 and 2 clinical trials are urgently needed for improved artemisinin-based combination therapies, and the C-17

steroid-artesunate conjugate holds promise to improve efficacy both against susceptible and resistant parasites.”

7) In the discussion, the authors should also discuss the tremendous potential for the steroid-linked primaquine to boost its anti-relapse activity against the dormant form (hypnozoites) of *P. vivax* in the liver. Some in vitro assays to assess this exquisite biology of *P. vivax* exist today.

Authors' response:

We are pleased that the reviewer sees additional potential for our approach and have included this aspect in the discussion (line 386): “It will be interesting to explore the potential for the steroid-linked primaquine conjugates in the therapy of other *Plasmodium* species and whether the conjugation approach might for example boost the anti-relapse activity against dormant liver stages (hypnozoites) of *Plasmodium vivax*.”

Minor points:

1) In the legend of figure 1, a spelling of DIC (differential interference contrast) is needed for readers who are not microscopists.

Authors' response:

We have addressed this oversight.

2) Line 257, authors wrote that emergence and spread of mutations in the parasite protein Kelch-13 results in delayed clearance meaning that there is growing concern

that treatment failures will develop. Authors should rephrase because today this is more than a growing concern. Few years ago, treatments with DHA-piperaquine were shown to lead to failure in Cambodia due to partial artemisinin resistance and resistance to piperaquine.

Authors' response:

Addressed. The section reads now: “Artemisinin combination therapies (ACTs) are the main antimalarial therapy in many parts of the world and often the drug of last resort, but emergence and spread of mutations in the parasite protein Kelch-13 results in delayed clearance and clinical treatment failures due to these mutations have been reported (48–50). The management of ACT-resistant parasites requires longer treatment times, which impacts on cost and compliance with the treatment regime, and increases the chance of patient morbidity and mortality before the infection can be cured (48, 49).” (line 285ff).

3) Line 274-275. The authors wrote "We utilised two parasite strains: an isolate from Cambodia (CAM3.II) with a mutated Kelch-13 conferring resistance, and a genetically modified revertant of this strain (CAM3.IIREV) conferring drug susceptibility (52)". It would be good to have an indication by the authors of what the mutation is. The most widespread in Southeast Asia so far is the Kelch13-C580Y. For non-expert readers, this information will be important.

Authors' response:

Addressed – the mutation C580Y is now mentioned in the text (line 310).

4) Line 392 authors wrote: "...and so the delivery system might also relevant to these pathogens (66, 67)". This sentence should read: ...and so the delivery system might also be relevant to these pathogens (66, 67).

Authors' response:

This oversight has now been corrected.

10th Apr 2024

Dear Prof. Maier,

Thank you for the submission of your revised manuscript to EMBO Molecular Medicine. We have now heard back from the one referee who agreed to evaluate your manuscript. This referee also assessed author responses to concerns raised by other referee. I am pleased to inform you that we will be able to accept your manuscript pending the following final amendments:

- 1) Authors: We note a discrepancy of author's name: Blake Curtis in the manuscript and Blake Curtiis in our submission system. Please correct.
- 2) In the main manuscript file, please do the following:
 - Please address all comments suggested by our data editors listed below:
 - o Figure legends:
 1. Please define the annotated p values **/* in the legend of figure EV 1a; as appropriate.
 2. Please note that in figure 4c; there is a mismatch between the annotated p values in the figure legend and the annotated p values in the figure file that should be corrected.
 3. Please note that information related to n is missing in the legends of figures 2c; 3f; EV 1b-d.
 4. Please note that the error bars are not defined in the legend of figure 2c.
 - The manuscript sections should be in the following order: Title page - Abstract & Keywords - Introduction - Results - Discussion - Methods - Data Availability - Acknowledgments - Disclosure Statement & Competing Interests - References - Figure Legends - Expanded View Figure Legends.
 - Statistical paragraph should reflect all information that you have filled in the Authors Checklist, especially regarding randomization, blinding, replication etc.
 - Rename "Competing interests" to "Disclosure and competing interests statement". We updated our journal's competing interests policy in January 2022 and request authors to consider both actual and perceived competing interests. Please review the policy <https://www.embopress.org/competing-interests> and update your competing interests if necessary.
 - Author contributions: Please remove it from the manuscript and specify author contributions in our submission system. CRediT has replaced the traditional author contributions section because it offers a systematic machine-readable author contributions format that allows for more effective research assessment. You are encouraged to use the free text boxes beneath each contributing author's name to add specific details on the author's contribution. More information is available in our guide to authors:
<https://www.embopress.org/page/journal/17574684/authorguide#authorshipguidelines>
 - Rename "Data and Materials availability" to "Data availability". Remove the current sentence and add the following statement: This study includes no data deposited in external repositories.
 - Correct the reference citation in the text and reference list. In the text, a reference should be cited by author and year of publication. Include a space between a word and the opening parenthesis of the reference that follows. In the reference list, citations should be listed in alphabetical order. Where there are more than 10 authors on a paper, 10 will be listed, followed by "et al.". Please check "Author Guidelines" for more information.
<https://www.embopress.org/page/journal/17574684/authorguide#referencesformat>
- 3) Appendix: Please upload it as a PDF file and add title page with the table of content and page numbers. For example: Chemical synthesis - 1-24, Biological methods - 25-34 and References - 35-38.
- 4) Funding: Please merge it with "Acknowledgments". Also, make sure that information about all sources of funding are complete in both our submission system and in the manuscript. Currently, DP180103212 and APP1182369 are in our submission system, but they are missing in the manuscript file, while Australian National University is in the manuscript file, but is missing in the submission system.
- 5) The Paper Explained: Please add it to the main manuscript file.
- 6) Synopsis:
 - Synopsis image: Please provide a striking image or visual abstract as a high-resolution jpeg file 550 px-wide x (250-400)-px high to illustrate your article.
 - Please check your synopsis text and image before submission with your revised manuscript. Please be aware that in the proof stage minor corrections only are allowed (e.g., typos).
- 7) For more information: This space should be used to list relevant web links for further consultation by our readers. Could you identify some relevant ones and provide such information as well? Some examples are patient associations, relevant databases, OMIM/proteins/genes links, author's websites, etc...
- 8) Source data: My colleague Hannah Sonntag requested the source data on 04. 12.2023 and sent Source Data Checklist (attached to this letter). As source data are now mandatory for all manuscripts published in our journal, please provide all requested data and upload them as one folder per figure.
- 9) As part of the EMBO Publications transparent editorial process initiative (see our Editorial at <http://embomolmed.embopress.org/content/2/9/329>), EMBO Molecular Medicine will publish online a Review Process File (RPF) to accompany accepted manuscripts. This file will be published in conjunction with your paper and will include the anonymous referee reports, your point-by-point response and all pertinent correspondence relating to the manuscript. Let us know whether you agree with the publication of the RPF and as here, if you want to remove or not any figures from it prior to publication.

10) Please provide a point-by-point letter INCLUDING my comments as well as the reviewer's reports and your detailed responses (as Word file).

I look forward to reading a new revised version of your manuscript as soon as possible.

Yours sincerely,

Zeljko Durdevic

*** Instructions to submit your revised manuscript ***

1) a .docx formatted version of the manuscript text (including Figure legends and tables)

2) Separate figure files*

3) supplemental information as Expanded View and/or Appendix. Please carefully check the authors guidelines for formatting Expanded view and Appendix figures and tables at <https://www.embopress.org/page/journal/17574684/authorguide#expandedview>

4) a letter INCLUDING the reviewer's reports and your detailed responses to their comments (as Word file).

5) The paper explained: EMBO Molecular Medicine articles are accompanied by a summary of the articles to emphasize the major findings in the paper and their medical implications for the non-specialist reader. Please provide a draft summary of your article highlighting

This may be edited to ensure that readers understand the significance and context of the research.

Please refer to any of our published articles for an example.

6) For more information: There is space at the end of each article to list relevant web links for further consultation by our readers. Could you identify some relevant ones and provide such information as well? Some examples are patient associations, relevant databases, OMIM/proteins/genes links, author's websites, etc...

7) Author contributions: the contribution of every author must be detailed in a separate section.

8) EMBO Molecular Medicine now requires a complete author checklist (<https://www.embopress.org/page/journal/17574684/authorguide>) to be submitted with all revised manuscripts. Please use the

checklist as guideline for the sort of information we need WITHIN the manuscript. The checklist should only be filled with page numbers where the information can be found. This is particularly important for animal reporting, antibody dilutions (missing) and exact values and n that should be indicated instead of a range.

9) Every published paper now includes a 'Synopsis' to further enhance discoverability. Synopses are displayed on the journal webpage and are freely accessible to all readers. They include a short stand first (maximum of 300 characters, including space) as well as 2-5 one sentence bullet points that summarise the paper. Please write the bullet points to summarise the key NEW findings. They should be designed to be complementary to the abstract - i.e. not repeat the same text. We encourage inclusion of key acronyms and quantitative information (maximum of 30 words / bullet point). Please use the passive voice. Please attach these in a separate file or send them by email, we will incorporate them accordingly.

You are also welcome to suggest a striking image or visual abstract to illustrate your article. If you do please provide a jpeg file 550 px-wide x 300-800px high.

10) A Conflict of Interest statement should be provided in the main text

11) Please note that we now mandate that all corresponding authors list an ORCID digital identifier. This takes <90 seconds to complete. We encourage all authors to supply an ORCID identifier, which will be linked to their name for unambiguous name identification.

Currently, our records indicate that the ORCID for your account is 0000-0001-7369-1058.

Please click the link below to modify this ORCID:
Link Not Available

Graphs 800-1,200 DPI
Photos 400-800 DPI
Colour (only CMYK) 300-400 DPI"

*Additional important information regarding figures and illustrations can be found at
<https://bit.ly/EMBOPressFigurePreparationGuideline>. See also figure legend preparation guidelines:
<https://www.embopress.org/page/journal/17574684/authorguide#figureformat>

***** Reviewer's comments *****

Referee #1 (Comments on Novelty/Model System for Author):

IN the absence of in vivo data, the clinical relevance of the findings obtained using the in vitro system employed in this study remains unclear.

Referee #1 (Remarks for Author):

While I remain of the opinion that the choice of Primquine for this proof-of-principle study is highly debatable, and that in vivo data would greatly increase the relevance of the authors' findings, I understand their arguments in both these regards.

Referee #2 (Remarks for Author):

I did review carefully point per point the answers of the authors and their corrections.
I agree with all of them, and consequently, I strongly support the publication of this revised manuscript in EMBO Molecular Medicine.

The authors addressed the minor editorial issues.

24th May 2024

Dear Prof. Maier,

We are pleased to inform you that your manuscript is accepted for publication and is now being sent to our publisher to be included in the next available issue of EMBO Molecular Medicine.
